# Gravity as a tool to improve the hydrologic mass budget in karstic areas

Tommaso Pivetta[1], Carla Braitenberg[1], Franci Gabrovšek[2], Gerald Gabriel[3, 4], Bruno Meurers[5]

[1] Department of Mathematics and Geosciences, University of Trieste, Trieste, 34128, Italy
[2] Karst Research Institute ZRC SAZU, Postojna, 6230, Slovenia
[3] Leibniz Institute for Applied Geophysics, Hannover, 30655, Germany
[4] Institute of Geology, Leibniz University Hannover, Hannover, 30167, Germany
[5] Department of Meteorology and Geophysics, University of Vienna, Wien, 1090, Austria

*Correspondence to*: Tommaso Pivetta (tommasopivetta@yahoo.it)

## Abstract

Monitoring the water movements in karstic areas is a fundamental but challenging task due to the complexity of the drainage system and the difficulty in deploying a network of observations. Gravimetry offers a valid complement to classical hydrologic measurements in order to characterize such systems in which the recharge process causes temporarily accumulation of large water volumes in the voids of the epi-phreatic system. We show an innovative integration of gravimetric and hydrologic observations that constrains a hydrodynamic model of the Škocjan cave system (Slovenia). We demonstrate how the inclusion of gravity observations improves water mass budget estimates for the Škocjan area based on hydrological observations only. Finally, the detectability of water storage variations in other karstic contexts is discussed with respect to the noise performances of spring and super-conducting gravimeters.

## 1 Introduction

Karst areas on carbonates and evaporates occupy about 15 % of the ice-free continents; about one fourth of the world's and about one third of Europe's population is supplied by water from karst aquifers. In such aquifers most of the water is drained through networks of solution conduits, which evolve along sedimentary or tectonic discontinuities (Ford and Williams, 2007). The evolution of conduit systems is controlled by complex mechanisms, which make the position of drainage pathways hard to predict. The evolution of karst aquifers is driven toward equilibrium, where conduit systems effectively drain all of the available recharge; however, in active tectonic environments, the evolution is being continuously stirred by changes of boundary conditions and structure (Gabrovšek et al., 2014). This leads to complex geometries of networks, with high variations of conduit cross-sections and abrupt terminations of channels by breakdowns or fault planes. Such systems are permanently out of equilibrium, and exhibit large water level variations during flood events. Large voids with volumes in the order of $10^6$

m$^3$ are common features in such settings. The positions of solution conduits and voids in karst aquifers are largely unknown, except for the parts accessible to direct human exploration. To assess the structure of karst aquifers and the response to the recharge process, different geophysical and hydrological techniques are used, each of them being applicable to a specific
situation.

Where groundwater flow is accessible through caves, classical hydrological instrumentation such as pressure, temperature and electrical conductivity loggers may be deployed. This instrumentation is indispensable to verify hydraulic connection between cave systems during a rain event and determine the travel time of the water masses or pollutants. Such data together with meteorological observations are fundamental to constrain hydrologic and hydraulic models.

Groundwater flow between accessible points (sinks, water caves, springs) can also be verified by tracing techniques and geochemical analysis (Petrič and Kogovšek, 2016; Zini et al., 2014; Goldscheider and Drew, 2014). However these techniques, do not give information on the flow distribution and aquifer structure between the observed points.

Geophysical methods are complementary tools to these more classical hydrological prospections (Chalikakis et al., 2011): alternation of rock and voids in combination with water level variations during the recharge process cause variations of the
physical parameters, such as seismic wave velocity, electrical resistivity and the electrical permeability. Mass variation associated to the water mass redistribution causes changes in the gravitational acceleration. Therefore, e.g., time-lapse electric resistivity measurements (Watlet et al., 2018), seismic ambient noise observations and gravimetry (Fores et al., 2018) can provide valuable information on the underground karstic structures, preferential water paths and water storage variations.

The sensitivity of each method to the recharge process depends on the depth at which the water variations occur. In addition
to this, the temporal and spatial scales of the underground flow variations as well as logistic considerations and the need of further data to correct any other non-hydrologic component should be considered carefully when designing a geophysical campaign. For instance, electrical resistivity tomography (ERT) can be used to characterize both shallow and deeper portions of the aquifer, but the depth of investigation depends mostly on the maximum electrodes spacing. In addition to this, ERT needs the implementation of petro-physical relations in order to properly invert the resistivity in terms of saturation changes.

Gravimetry being sensitive to spatial and temporal mass variations, is particularly apt for studying the karstic aquifers and in general the water mass redistribution during the hydrologic cycle. Recently, the advent of new and performing instruments such as the superconducting gravimeters (SG) have raised new interest into this method for monitoring water mass movements (Van Camp et al., 2017). Gravity measurements have been successfully applied to study the groundwater flow in other karstic environments (Fores et al., 2017; Jacob et al., 2011; Meurers et al., 2021; Mouyen et al., 2019; Van Camp et al., 2006; Watlet
et al., 2020) or to monitor the subsidence and underground mass redistribution in sinkhole prone areas (Kobe et al., 2019). Other studies utilized gravimetry to characterize water flow in porous media (Güntner et al., 2017; Weise and Jahr, 2017) or to monitor fluids relevant for geothermal exploitation (Hinderer et al., 2016; Portier et al., 2018).

Compared to the other geophysical techniques for monitoring ground water movements, gravimetry requires less petro-physical relations, as the change in gravity is simply related to the water density that is frequently assumed as constant in time
and space. This aspect clearly simplifies the interpretation of the observed gravity transients. Another strength of the method

is related to its sensitivity to the integrated water mass around the instrument, which implies that a remote monitoring of the water storage unit is possible. This is an important aspect that allows to fill the gaps of the sparse water head observations, which depend on the accessibility of the caves in the vadose zone. Most of the gravity signal originates from the mass variations occurring just below the instrument, however the horizontal sensitivity increases as the storage unit is deeper (Van Camp et al., 2017). Being sensitive to the integrated water mass is also a key aspect to obtain a reliable mass flux balance of the system complementing the head observations, which usually are representative of a very localized portion of the system.

However, since gravimetry belongs to the potential field methods it suffers from the non-uniqueness of the inverse problem: in other words, we can't assign a unique density distribution to a given gravity observation, hence, an infinite number of different underground mass distributions can in principle explain the observed gravity variations. Moreover, several other processes apart from the hydrology superpose on the gravity observations, requiring meticulous processing to isolate the component of interest. Largest components are due to Earth and marine tides and atmospheric effects, which, fortunately, can be accounted for with sufficient precision (Van Camp et al., 2017; Watlet et al., 2020). In any case, areas close to the sea pose some further challenges for such corrections and the supplement uncertainties should be taken into consideration when interpreting the hydrologic related signals.

The non-uniqueness of the inverse problem clearly affects the interpretation of the gravity observations and requires implementing hydrologic models that simulate the recharge process and the sequent groundwater time evolution. These models often assume that the karstic drainage system, although being a complex medium, can be reasonably well approximated by a porous media in which the water is homogeneously stored in the matrix porosity. The approximation seems valid in several contexts, mostly with autogenic recharge, where a diffuse network of sub-superficial fractures that drains the waters into a more organized channel system in the vadose zone is present. An example of these applications can be found in Fores et al. (2017), who firstly inverted the gravity variations observed by an SG in the Larzac plateau in terms of equivalent water height (EWH) through the simple Bouguer formula. Then the authors modelled the EWH time-series employing two tanks connected in series, governed by the Maillet law (Deville et al., 2013), which simulate two karst reservoirs. In their interpretation, the first tank is associated to a very shallow soil layer while the second represents the epikarst. The usefulness of implementing the tank model is that they can simulate also complex non-linear effects (as the piston effect) and obtain tank parameters, such as characteristic transfer time of the reservoirs, which can be linked to the groundwater velocity of the infiltration water. Similar applications that assume the karst to be approximated by a porous media can be found in Jacob et al. (2008), Deville et al. (2013) or Mouyen et al. (2019).

Some efforts in taking into account the complex network of conduits and fractures in a karstic aquifer and the sequent inhomogeneous distribution of the water masses during the recharge process have been proposed by Jacob (2010) and recently by Watlet et al. (2020) with an application of gravimetry in the Rochefort caves in Belgium. In the Rochefort cave, the water volume is influenced by the allogenic contribution provided by the Lomme River. The authors forward modelled the gravity effects of the water level variations in the known cavities, using the diver data as constraint. The authors showed that the observations are up to 8 times higher with respect to the model outcome, suggesting the presence of further unknown storage

units in the vicinity, probably fed by the autogenic contribution. Apart from these last two contributions, taking into account the heterogeneous distribution is challenging since it requires information on spatial distribution of voids and conduits and their interconnectivity. In our study, we make use of the known 3D geometry of a cave system to interpret the gravity observations by implementing a 3D hydraulic model.

In an allogenic context, the main ground water paths are usually more easily discernible and frequently follow a strongly
channelized system that can hardly be represented by a homogeneous media. An example is the Classical Karst, which is the focus of the present study. The Classical Karst hydrodynamics is dominated by the highly variable allogenic recharge of the Reka River, which causes large water level and storage variations in the epiphreatic zone. The initial part of the Reka River underground flow is formed by the Škocjan Caves, where flow follows a large underground canyon with a total volume of more than $5 \cdot 10^6$ m$^3$ and fast water level variations ($> 100$ m) during large flood events of the river. Then the Reka River
continues its underground flow in several other caves finally reaching the Adriatic Sea about 30 km North East from the Škocjan Caves. As shown by Gabrovšek et al. (2018), the water level response of such underground karstic system to the discharge variations of the Reka can be predicted using the same equations used for modelling the river hydrodynamics (i.e. Saint Venant equations). In their work, the authors discretized the underground voids through a series of conduits with elementary cross sections and the water level variations in the conduit system are predicted supplying the observed river
discharge as input. The water head observations from six divers located in the caves intersecting the underground water path are used to adjust iteratively their hydraulic model. Similar applications can be found for studying other karstic contexts as the Postojna caves (Kaufmann et al., 2016) or the Planinsko Polje (Mayaud et al., 2019). The final model of Gabrovšek et al. (2018) is relevant for several scopes, in particular for understanding the complex mechanism of water accumulation in the voids during the recharge process, which is highly non-linear, for estimating the water transfer velocity from the different cave
systems and for formulating a mass balance of the karst system. The lack of head observations in several points introduces uncertainties in their model, which is fairly well constrained at the beginning of the Reka underground path, becoming more uncertain going downstream where also the vadose zone is deeper and hardly accessible. An ideal complement in these portions of the aquifer would be provided by remote and indirect observations of the hydrologic behavior and gravimetry is an ideal technique to further constrain the model.

In light of these considerations, in July 2018 a continuously recording spring-based gravimeter was installed in the Classical Karst area, nearby the Škocjan cave system, which is still operating. Given the exceptional dimensions and the large hydrologic variations observed, the Škocjan cave system represents an ideal site to evaluate the combined gravimetric-hydrological response to the Reka flood events and to test demands on these hydraulic models in terms of allowed simplifications. Up to now about 2 years of data have been analyzed and several flood events, including an extreme flood in February 2019, provided
excellent gravimetric and hydrologic records that we aim to model and explain in the frame of this study.

Our work contributes in demonstrating the usefulness of gravity measurements for hydrologic studies; in particular, we show that:

1) fast hydrologic variations in a cave system in the vadose zone can be effectively monitored by employing one spring-based gravimeter;

2) integration of hydraulic models, head observations and gravity helps in better assessing the water mass balance of the system;

3) geophysical effects superposing on the hydrologic gravity contributions can be discerned and removed even in challenging areas as the Classical Karst, which is nearby the Adriatic Sea.

We finally also give a wider perspective of the feasibility of such an approach to study the structure of more general karst
aquifers and groundwater variations therein.

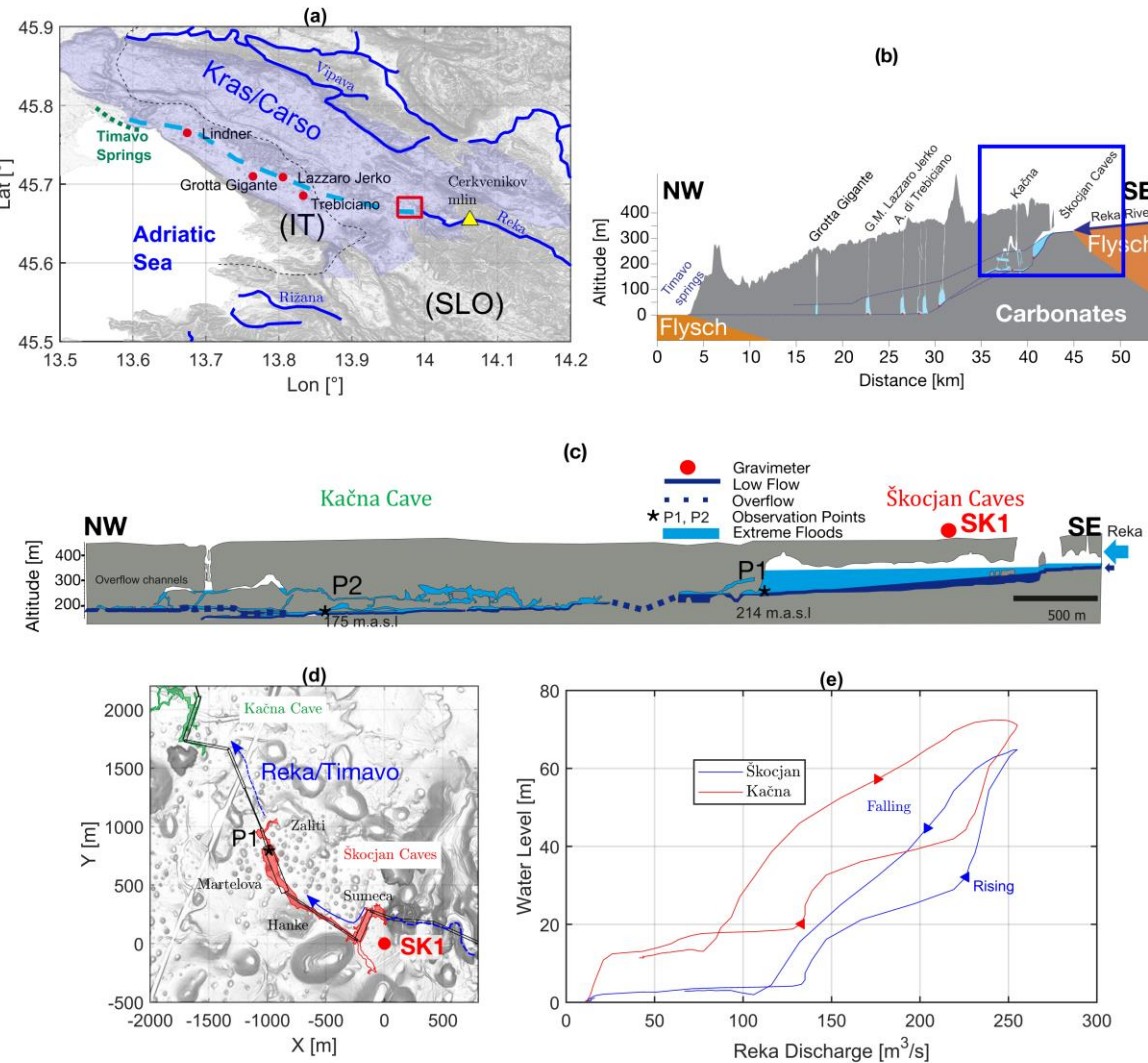

**Figure 1 Hydrologic characteristics of the Kras/Carso and Škocjan area. a) Hillshade map of the Kras/Carso showing the Škocjan caves area (red box) and other main caves (red dots) along the Reka underground track (cyan dashed line). The yellow triangle shows the Cerkvenikov Mlin location, a monitoring station of the Reka. The transparent blue area shows the borders of the limestone formations (Jurkovšek et al., 2016) in the Kras/Carso area. Black stippled line: national borders. The Digital Elevation Model (DEM) is derived from the SRTM mission and has a resolution of 30 m (Farr et al., 2007). b) Geological cross-section along the Reka-Timavo system (cyan dotted line in a) showing the main cave systems. Blue box outlines the Škocjan and Kačnab systems which are shown in detail in plot c). d) Hillshade of**

the Škocjan area (red box in plot a) from laser-scan superposed on the outlines of Škocjan (red line) and Kačna cave systems (green line). The red areas show the portions of the Škocjan caves where water is stored during extreme events. Names of the channels and halls are reported for Škocjan. Red dot: location of the gravimeter. Black star: location of the diver (P1) Black lines show the plan view of the hydraulic model discussed in section 3. The laser-scan is available from http://gis.arso.gov.si/evode/profile.aspx?id=atlas_voda_Lidar@Arso&culture=en-US. e) Stage vs. discharge of Reka responses to an extreme flood event, occurred on 12/12/2008 lasting for 1.5 days and recorded at Škocjan and Kačna.

## 2 Research area

The Classical Karst (Kras in Slovene/Carso in Italian) plateau between Slovenia and Italy is about 40 km long and in average 13 km wide and it is underlain by a up to several kilometres thick succession of Cretaceous to Lower Paleogene carbonates (Jurkovšek et al., 2016). The geological structure of the broader area is a result of the collision between the Apulian and Eurasian lithospheric plates. The Kras/Carso Plateau is an anticlinorium, which structurally belongs to the External Dinaric Imbricated Belt (Moulin et al., 2016). The carbonates are surrounded by relatively impermeable flysch with fluvial network providing allogenic recharge to karst. Flysch also prevents the outflow along the SW boundary, so that the main flow is forced to follow the Dinaric (SE–NW) direction (cyan dashed line in Figure 1a). Along the NW coast of the Trieste Bay, the limestone flysch contact is located systematically below the sea level allowing the outflow of the karst waters into the sea through numerous springs, located both above and below the sea level. Among these, the Timavo springs with average discharge of 30 $m^3$/s are the most important.

The recharge of the system is manifold: beside diffuse autogenic infiltration from the karst surface, groundwater inflow from the adjacent alluvial aquifer is a dominant recharge contributor at low flow. During floods, the main contribution comes from the Reka River, which sinks at the Škocjan Caves, near the flysch-limestone contact on the SE border of the plateau (Figure 1a). The straight line of its underground flow between the Škocjan Caves and the Timavo Springs is about 33 km. The long-term (1952–2013) average discharge is about 8 $m^3$/s (ARSO, 2016). The ratio between the highest and the lowest flow rate is ~1700, with the maximum measured discharge 305 $m^3$/s. Since the climate in the area is transitional between Mediterranean and Continental most of the rain events occur in autumn. Annual precipitation in the recharge area reaches over 2000 mm with single precipitation events that exceed 250 mm in 12 h (Gabrovšek et al., 2018). Usually such rains are associated with large discharge variations of Reka, exceeding 250 $m^3$/s and lasting for several hours. The occurrence of these events is about every 1.5-2 years, while more common seasonal precipitations cause events with peak discharge in the range of 80-150 $m^3$/s.

The groundwater flow in the aquifer of Kras/Carso is characterised by highly variable recharge of the Reka River and an irregular structure of conduits and voids. The thickness of the vadose zone ranges from several tens of meters close to the

Timavo springs, to over 300 m at the SE border. The caves, that evolved over 5 Ma, populate the entire vadose zone (Figure 1b).

In this study we focus on caves of the epi-phreatic (also flood level) zone. Practically all known epi-phreatic caves of the Kras/Carso aquifer have large voids above the base flow water table and experience high and rapid rise of the water level during floods. These caves present major temporal storages of floodwater. Most of them have been discovered by following

and excavating passages along the air-flow which is pushed out from the epi-phreatic passages and voids by the rising water. Many blowholes on the surface with non-accessible connection to the epi-phreatic level indicate existence of yet unknown large voids.

Groundwater level, temperature, and specific electric conductivity in the epi-phreatic caves are monitored by autonomous instruments (Zini et al., 2014; Gabrovšek et al., 2018). The piezometric recordings have revealed water level changes reaching

120 m with rates up to 10 m/h. Considering that the planar area of some known large chambers reaches 10,000 m$^2$, the local rate of change of mass is in the order of $10^8$ kg/h.

Groundwater dynamics was also monitored by geodetic observations with tiltmeters and GNSS placed in Grotta Gigante (Braitenberg et al., 2019), which evidenced the presence of overpressure in conduits below the cave.

**3 Hydrology and monitoring network of the Škocjan/Kačna system**

The underground flow path of the Reka River starts in Škocjan Caves, where the flow follows a large underground canyon for more than 3 km (Figure 1c). The cross-section of the canyon is between 2,000 m$^2$ and 12,000 m$^2$, as determined from the internal topographic surveys conducted in the past years inside the cave system (see Appendix B for details). Along several sections, the canyon widens into large chambers; Martelova chamber being the last and the largest with a volume of $2.6 \cdot 10^6$

m$^3$. It is terminated by a fault plane, where the channel cross-section diminishes to few tens of square meters. The hypothesis is that the active fault intersected the solution channel reducing its cross-section. The fault movement counteracts the solution action of the water flow leading to abrupt cross section area changes.

The flow then follows a system of channels and flooded conduits, which continues in a yet unexplored flooded conduit, and reappears about 800 m NW in a sump in the Kačna Cave. The Kačna Cave is an over 13 km long network of channels, located

between the permanent flow level at 150-180 m a.s.l. and the surface of Kras plateau at about 430 m a.s.l. The permanent flow level is a 5 km long system of channels with variable geometry, interrupted by fully flooded conduits, which are still under exploration. A large part of the flow from Kačna Cave reappears in Kanjaduce Cave, about 6 km NE from Kačna Cave (Blatnik et al., 2020).

The long-term monitoring of groundwater parameters in most caves in Kras/Carso aquifer also includes temperature/pressure

loggers in Martelova chamber in Škocjan Caves (P1 on Figure 1c) and in Kačna Cave (P2). Recharge to the system is given

by the records of the gauging station located 5 km upstream from the Škocjan Caves (maintained by the Slovenian Environmental Agency, ARSO; Figure 1a). Table 1 reports the data availability for the different sensors employed in the study.

| Sensor name | Measurement | Data availability intervals (month/year) | Temporal resolution |
|---|---|---|---|
| P1 | Diver for temperature and pressure | 1/2005-1/2008; 6/2008-6/2009; 1/2018-ongoing | 1 hour |
| P2 | Diver for temperature and pressure | 1/2005-1/2007; 1/2008-6/2011; 1/2013-6/2014 | 1 hour |
| SK1 | Gravity variations | 7/2018-ongoing | 1 second |
| Cerkvenikov Mlin | Discharge measurement | 1/1952-ongoing | 30 minutes |

**Table 1 Hydrologic and geophysical data availability in the Škocjan area.**

Based on these data and speleological observations, Gabrovšek et al. (2018) developed a hydraulic model to describe the flood propagation over the whole epi-phreatic system of Reka. The relatively well-defined geometry of Škocjan and Kačna Caves allowed a well constrained model and a description of mechanisms leading to the observed water level dynamics in this part 220 of the system. The major inflection in the stage-recharge curves in both caves happens simultaneously, when the discharge of Reka reaches 130 $m^3$/s. Afterwards the level in both caves rises rapidly with increasing recharge, with several inflections (Figure 1e). It can rise above 100 m when the discharge is larger than 300 $m^3$/s with rate of increase of up to 10 ($m^3$/s)/h. Gabrovšek et al (2018) showed that the Škocjan-Kačna system is back-flooded due to a flow restriction down-flow from the Kačna Cave. Back flood occurs in a portion of the system when the incoming flow from Reka river is greater than the discharge 225 flow of the conduits downstream; this leads to an increase of the water level in the caves.

As already hinted the extreme flood events, characterized by peak discharges of Reka over 250 $m^3$/s, have a recurrence time > 1.5 years, while typically at least one event per year is expected in the range of 150-200 $m^3$/s. During medium-low flow conditions, water level variations in the Martelova stay below 5 m.

To explore the gravity effect of the floods and to evaluate the accuracy of hydraulic models, a gPhone gravimeter (herein 230 referred as SK1) was installed on the surface above Škocjan Caves in July 2018. The gPhone gravimeter employs the zero-length spring together with a feedback system to perform continuous measurement of the temporal variations of the gravity acceleration; according to in-house specifications, the spring design assures a precision of about 10 nm/$s^2$ and drift rates lower than 5 $10^3$ nm/$s^2$/month. The meter is installed in a closed room, in a rarely accessed building, near the information centre (red dot in Figure 1d), approximately 250 m far from the Šumeča Chamber. It is equipped with a tripod, which allows for automatic 235 tilt compensation, and pressure and temperature sensors; all the data are recorded with a 1 Hz sampling rate. Up to now we have analysed about 2 years of data, which include 4 flood events with peak discharge < 200 $m^3$/s and an extraordinary one, which is discussed in detail in this paper and exceeded 290 $m^3$/s of peak discharge.

## 4 Reka flood event: joint gravity and hydraulic model

Between February 1$^{st}$ and February 3$^{rd}$ 2019, rainfall in the catchment area of Reka River amounted to more than 300 mm,
giving rise to a flood event with peak discharge of over 290 m$^3$/s (Figure 2a). The event was recorded by the instruments measuring water level and temperature in Martelova (P1) and by the gravimeter (SK1). The peak level in Martelova was about 91 m above the base flow.

In the following, we first present the gravity observations and the processing for extracting the residuals, i.e. the gravity variations caused solely by the flood event; then a joint hydraulic and gravimetric model of the flooding event is discussed.
The model has been iteratively adjusted in order to minimize the differences between observed and modelled gravity and hydrological data.

We finally demonstrate how the inclusion of the gravimetric model helps in better constraining the water mass fluxes in the cave.

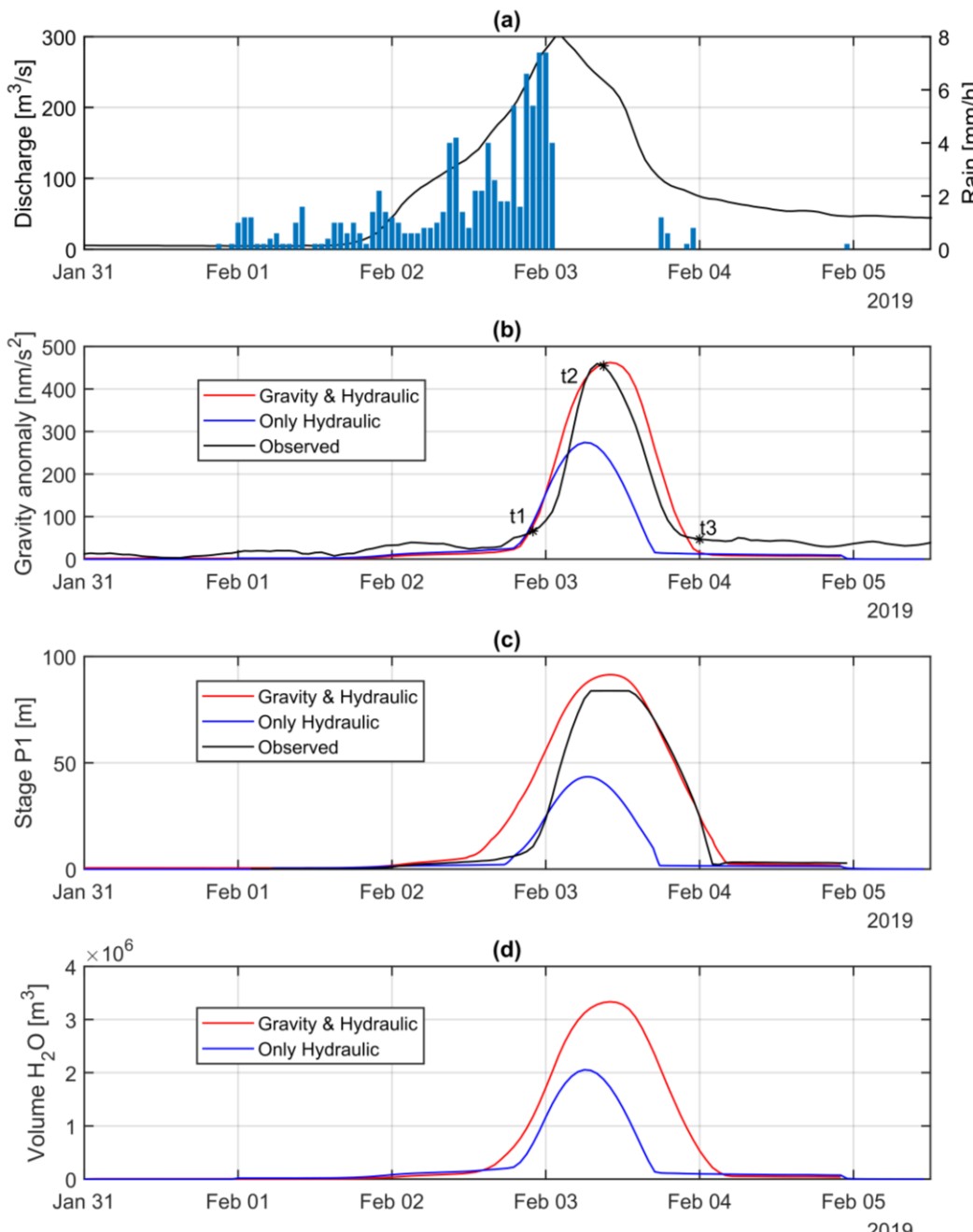

**Figure 2 Observed and modelled time series of the flood event in 02/2019 – red curves: new model considering gravity and hydraulic data, blue curves: former model only considering hydraulic data (Gabrovšek et al., 2018). a) Hydrological data: rain (blue bars) recorded in Škocjan and discharge (black line) at Cerkvenikov mlin. b) Observed and modelled gravity signals (black and red/blue) at SK1; t1, t2 and t3 indicate three different phases of the flood event**

**(rising, peak, falling). c) Stage time-series recorded and modelled at Martelova/P1 (black and red/blue). d) Water**
**volume accumulation during the event in the whole Škocjan system according to the new model (red) and based on Gabrovšek et al. (2018; blue line).**

### 4.1 Gravity observations and noise level assessment

The 2019 flood event generated a transient gravity signal of more than 400 nm/s$^2$ as shown in Figure 2b (black line) in terms
of the gravity residuals after the subtraction of gravity contributions due to tides, atmospheric pressure changes, and global hydrology (for any details of the comprehensive data analyses and processing we refer to Appendix A).

A local tide model, including Earth and ocean tidal effects, has been calculated by analysing a 2 years long gravity time series at Škocjan with the ETERNA-ET34-X-V73 package (Wenzel, 1996; Schüller, 2015). We retrieved local tidal parameters together with an empirical air pressure admittance in the diurnal and sub-diurnal frequency band. The admittance is defined as
the gravity change due to a 1hPa pressure perturbation; it is calculated by fitting a linear relation between the gravity time-series reduced for the tidal component and the pressure time-series. As detailed in the Appendix A, the value we found (-3.39 nm/s$^2$/hPa) is in good agreement with the theoretical predictions. For long-periodic tides beyond the diurnal band we used parameters that were derived from the theoretical body tide model by Dehant et al. (1999). We calculated the effects of pole motion and length of day variations based on Earth rotation data provided by the International Earth Rotation and Reference
System Service. After subtracting the tides and Earth rotation effects, the residuals have been further corrected for local and global air pressure variations, by combining the air pressure admittance with the 4D Atmacs model corrections (Klügel and Wziontek, 2009). The Newtonian and loading effects of global soil moisture variations have been removed using the mGlobe package (Mikolaj et al., 2016), relying on the GLDAS model (Rodell et al., 2004). By global we mean the gravity effect of the complete continental water masses excluding the area within a radius of 0.1° around the station.

The final residuals, i.e. the observed gravity changes of interest, can still contain gravity changes related to other sources than the Škocjan hydrology such as the non-tidal ocean (NTOL) contribution and the hydrologic contribution of the immediate surrounding of the station. As detailed in the supporting information, the largest contribution is due to NTOL, for which we expect signals lower than 10 nm/s$^2$ during the period of interest. The local hydrologic effect is even lower due to the shielding effect of the buildings and it is probably below 0.025 nm/s$^2$/mm.

The drift of the instrument appears large and highly non-linear as expected from a spring based gravimeter; for the time span July 2018-August 2019 the drift followed a logarithmic-like trend which was taken into account by fitting a 5$^{th}$ degree polynomial curve. The first 3-months of observations confirmed the in-house specifications for the drift (around $5 \cdot 10^3$ nm/s$^2$/month); from October the drift was even lower, with values around $10^3$ nm/s$^2$/month. Up to now, we do not have at disposal any absolute gravity measurements to check properly temporal variations of the drift and the presence of eventual
long-period signals superposing on it. In any case, for the scopes of our study, exactly estimating the drift is not crucial since we are focusing on fast water mass variations that typically last for about 1-2 days when we expect the drift to be mostly linear.

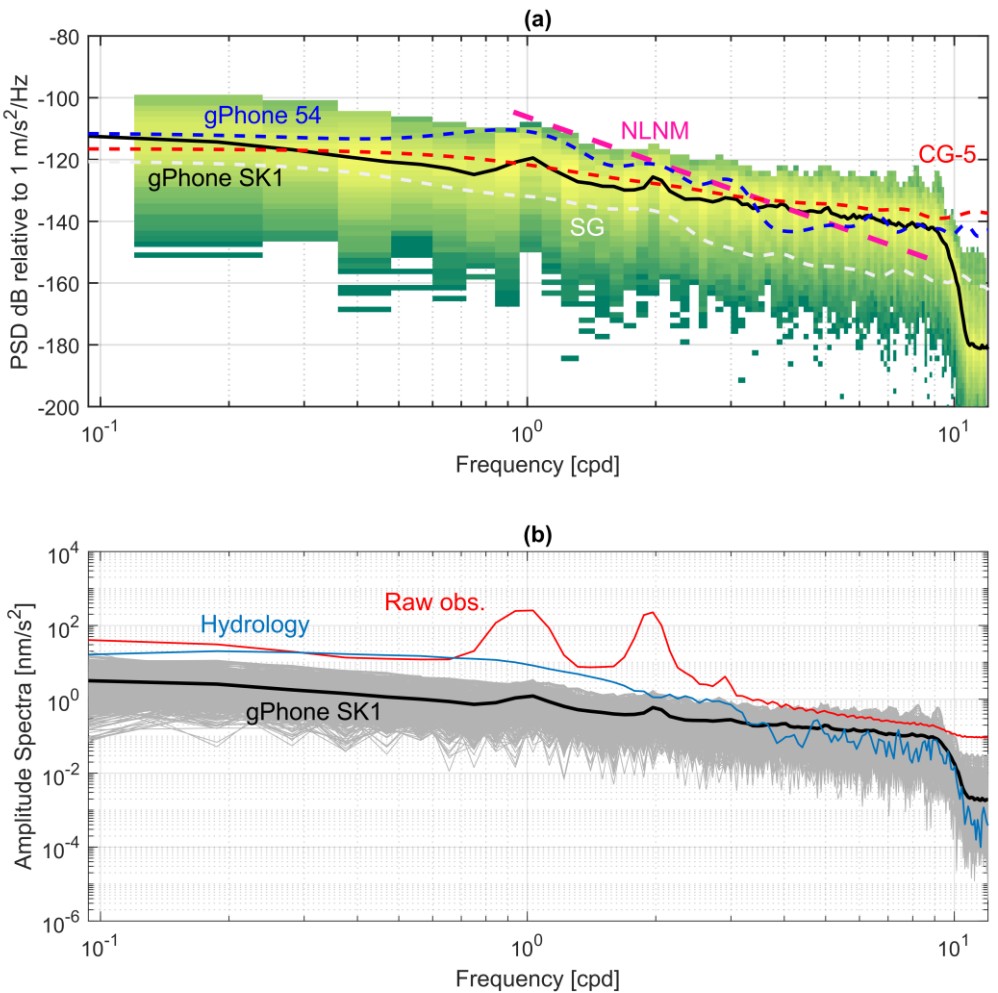

**Figure 3 Power Spectral Density (PSD) of the Škocjan gravity residuals. Frequency is expressed in cycles per day (cpd). Coloured patches: histogram showing the distribution of the PSDs; color code proportional to the number of PSD values for given intervals of frequency and dB. Black solid line: median PSD. For comparison median PSD of gravimeters installed in a Strasbourg station ((Riccardi et al., 2011)): blue dashed line: gPhone #54 ; red dashed line: PSD of the Scintrex CG5; white dashed line: PSD of the Superconducting gravimeter SG-C026. Purple thick dashed line: New Low Noise Model according to (Petersen, 1993). b) Amplitude spectrum of the Škocjan gravity residuals. Grey lines: individual spectral estimates. Black Solid line: median value. Blue curve: amplitude spectrum of the flood event in February 2019; red line: average amplitude spectrum for observations including tidal contribution and atmospheric effects.**

From the reduced time-series we estimated the noise level of the station, following the approach described in Rosat et al. (2017). In particular, we estimated the Power Spectral Density (PSD) by computing the Fourier transform (FT) on 10-days sliding windows, considering 1 hour resolution data. Before performing the FT, each window was tapered with a Hann window and the linear trends were removed by least squares fitting. We discarded periods when particularly strong hydrologic signals were recorded. For each frequency interval, we obtained a distribution of the PSDs from which the median value was computed and taken as reference for noise level.

In Figure 3a the distribution of the PSDs is reported with a 2D histogram: each bin of the histogram (a coloured patch) covers a frequency interval of 0.12 cycles per day (cpd) and 2 dB interval along the PSD axis; the colour is proportional to the number of spectrograms inside the bin. The median of the distribution is reported with the thick black line. For comparison we reported the New Low Noise Model (NLNM; Petersen, 1993) and also the noise curves of three gravimeters, the gPhone #54, the SG-C026 and the Scintrex CG-5, all installed in Strasbourg (Riccardi et al. 2011).

The SK1 station seems to perform better with respect to the noise level of the gPhone #54, which is noisier by about 10 dB in the diurnal and semi-diurnal band. This improvement is likely attributable to the automatic tilt compensation control which eliminates the tilt signals that deteriorate the quality of observations, as also previously pointed out by Watlet et al. (2020). In comparison with the Scintrex CG-5, SK1 has slightly lower noise levels in the 0.5-3.0 cpd frequency band while slightly higher noise is observed at lower frequency. The best performance is obviously achieved by the SG, which is systematically lower in comparison with all the other curves by at least an order of magnitude.

From the Figure 3b, which reports the amplitude spectra, we see that the SK1 noise level in the diurnal band is slightly above 1 $nm/s^2$ and is even lower (down to 0.3 $nm/s^2$) in the semi-diurnal band. In the same figure, we can also compare the noise level with the amplitude spectrum of the 2019 hydrologic event; in this case the signal to noise ratio (SNR) is about 20-25.

## 4.2 Hydraulic and gravimetric model

To validate the relation between the hydraulic and gravimetric signal, we have modelled both responses. As already hinted, a hydraulic model of the Škocjan-Kačna system was first presented by Gabrovšek et al. (2018) who demonstrated that the hydrodynamic response of a karstic system subjected to flow variations can be modelled reasonably well with simplified versions of the Navier-Stokes equations, the so-called Saint-Venant equations (SVe). With respect to the Navier-Stokes equations, the SVe model only 1-D flow, accounting the viscosity, boundary friction and turbulence terms through simple empirical relations (Blatnik et al., 2020). Due to their simplicity such equations are particularly apt to model unsteady flow occurring in artificial channels and also in natural rivers.

The SVe are derived from the mass and momentum conservation of a small piece of fluid subjected to an external force. Gabrovšek et al. (2018) used the implementation of the solution of the SVe in the Storm Water Management Model (SWMM) software developed by the Environmental Protection Agency (Rossman, 2017). SWMM is an open source and versatile

environment which was primarily designed to simulate urban sewage systems, but has been successfully applied to model conduit dominated karst systems. SWMM accounts for transitions between open channel and pressurised flow, and allows building complex conduit networks with arbitrary cross-sections. Storage is attributed to the volume of water in the conduits. The input into the Škocjan-Kačna system flow was given by the discharge hydrograph of the Reka River gauging station. The

geometry of the conduit system in the model was obtained from cave surveys; however, the modelling domain is highly simplified compared to reality. As shown by Gabrovšek et al. (2018) the flood behaviour of the system is strongly controlled by a limited number of flow restricting conduits; furthermore, the authors demonstrated that the modelling domain should include the Kačna cave system that highly influences the water dynamics in the Škocjan caves. The model used in the present study is an extension of the Gabrovšek et al. (2018) model in which a more realistic representation of the geometry of the

Škocjan caves is used. Our model, in particular, increases the number of conduits in the Škocjan cave from 2 to 4 and now includes a detailed discretization of Hanke and Sumeča channels (Figure 1d). The whole system is discretized by conduits with rectangular cross-section; the plan view of our model is shown in Figure 1d with the black lines.

We tested a more complex model that accounts for more realistic variations of the shapes of the conduits for the Škocjan caves; the results are detailed in the Appendix B and are in very good agreement with those discussed hereafter.

The full dynamic wave solution of the SVe equations was used, which allows to account for the backwater and reverse flow effects. Using the observed recharge value of the 2019 flood (Figure 2a), we modelled the time series of water levels in Martelova and we compared them with the measured ones (Figure 2c). The observed data show a large plateau at the peak flood, which is due to the fact that the water level exceeded the measurement range of the diver. In general, the model fits well to the observed data; the main difference is that the onset of the flood in the model precedes the recorded one. This time

difference is attributable to imperfect modelling of the Kačna cave for which the more complex water circulation is less known than for Škocjan. The final RMS difference between the observed and modelled water level data amounts to 8 m for the time span considered in Figure 2c.

Given the geometry of the conduits, their location in space and the simulated time-series of the water heights in the conduits, we have all the necessary information to build a 4D model of the mass variations below the gravimeter. The correct

interpretation of the gravity signal induced by hydrology requires calculating the effect of the underground mass variation induced by the water flows. The model should reproduce as accurately as possible the mass distribution, including a correct spatial location of the channels which is not required for fitting the hydraulic observations.

We discretized the water mass distribution occurring in the conduits at each time-step of the simulation through a series of small prisms, for which the gravity effect is calculated analytically at SK1. The base area of each prism is set to be 5 m $\times$ 5 m,

while the height is equal to the simulated water level in the conduit. To each prism, a density of 1000 kg/m$^3$ is assigned. Figure 2b shows the comparison between the observed gravity residuals and the model predictions. The model over-estimates the duration of the flood event by less than 5 hours, while the magnitude and the overall shape are well reproduced; the RMS error amounts to 40 nm/s$^2$ for the time span considered in Figure 2b.

The plots of Figure 4 show three snapshots of the simulated water levels in the caves during different phases of the flood. At
the time t1, the discharge of the Reka is increasing and the high flow conditions have already caused the flooding of the Kačna
cave. A back-flooding wave starts to propagate towards Škocjan; however in Martelova and in the other chambers the flow is
still mostly in an open channel regime. When new water input from the Reka is provided, the back flood becomes more
vigorous and starts affecting the Škocjan caves: now the water level in the caves rapidly raises at rates exceeding 10 m/h. The
following time-step t2 shows the situation when the Reka discharge has already reached the peak and it is on the falling limb
(about 250 m$^3$/s), but the back flood is still sustained and the water level in the caves has just reached the maximum as
observable in both P1 and SK1 plots (Figures 2b-2c). During this phase most of the gravity signal originates from the Sumeča,
as observable in Figure 5a, which is the closest cave to the gravimeter, while Hanke and Martelova contribute respectively
with 50 nm/s$^2$ and 10 nm/s$^2$. The remainder part of the signal is attributable to water mass accumulation outside the cave
system, in particular from water level variations of the Reka occurring in the two collapsed dolines located before the entrance
of the cave system.

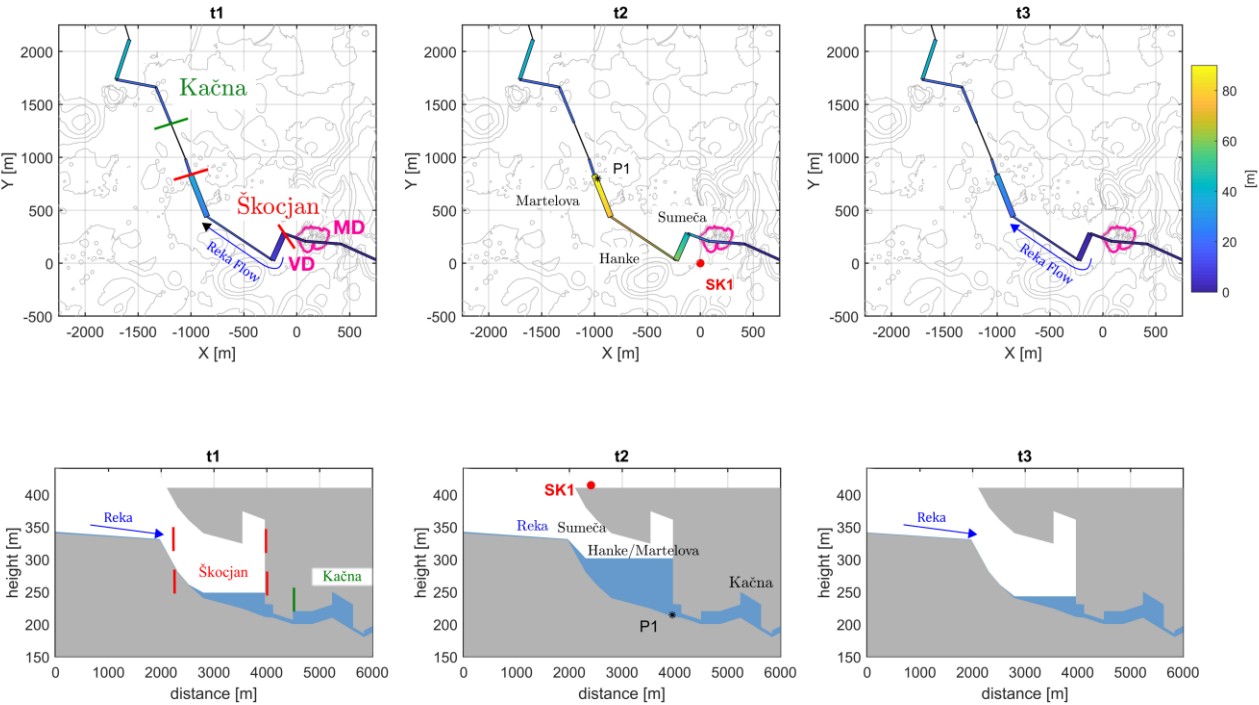

**Figure 4 Water distribution in the caves at three snapshots t1, t2, t3 (refer to Figure 2b): for each time-step, a cross
section and a plan view of the water levels in the caves are displayed. Note that in the cross sections the vertical scale is
exaggerated by a factor of about 10 and that the cross sections are plotted from east to west. The colour code is**

**proportional to the water level height in the conduits. The contours of the DEM are shown in grey. The origin of the reference system is centred on the gravimeter SK1 (red dot in snapshot t2). The cross section follows the path of the channel system, marked in the plan view. The distance is referred to the first node approximating the Cerkvenikov Mlin position. In the t1 snapshots the limits of the Skocjan cave system is displayed through the red lines; beginning of Kačna cave system in green. SK1: gravimeter; P1: diver in Martelova. The coalescent dolines, Mala Dolina (MD) and**

**Velika Dolina (VD) are shown with the purple outline.**

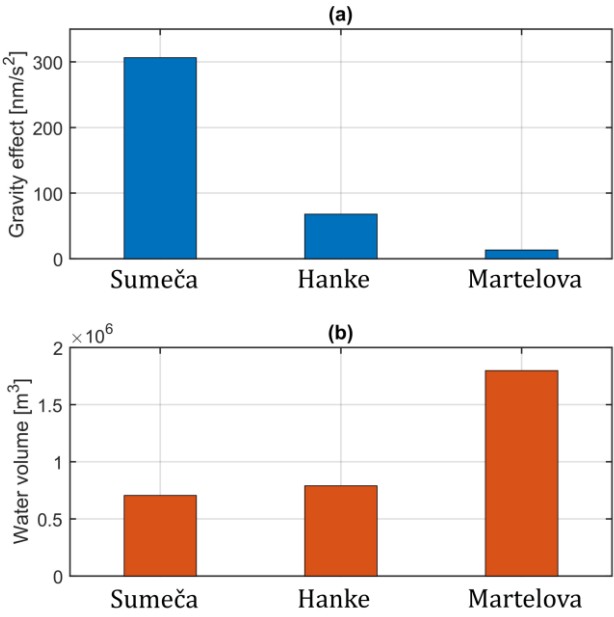

**Figure 5 Gravity contribution at SK1 and water volumes at the flood's peak.  a) Gravity contribution of each of the three chambers at the flooding peak (t2; in Figure 4). b) Stored water volume in each chamber at the flooding peak.**

The last snapshot (t3) depicts the situation during low flow conditions, when Martelova and the drainage system towards Kačna is slowly emptying. By comparing these three snapshots we observe the different storage periods in Sumeča/Hanke chambers with respect to Martelova; this is also testified by the differences in the duration of the transients recorded in P1 and SK1. After t3 the water level in the cave system has reached almost the base level while, interestingly, the gravity seems to be slightly higher with respect to the situation before the flood event (Figure 2b). Similar situation has been observed in the

Rochefort karst system (Watlet et al., 2020) and has been interpreted as due to increase in water content in the vadose zone due to delayed release of infiltrated waters. In our case the offset, calculated as the difference between the average gravity before and after the 2019 flood event, is about 20 nm/s$^2$.

### 4.3 Water volume balance and uncertainty assessment

Our hydraulic model predicts an accumulation of over 3.5 million m³ of water in the whole cave system at peak's flood (t2):
most of the water volume, about $1.8 \cdot 10^6$ m³, is stored in the Martelova (Figure 5e) while Hanke and Sumeča chambers represent smaller storage volumes ($0.7 \cdot 10^6$ m³ each one).

The total volume stored in the caves predicted by our new model is almost 30% bigger compared to the estimates of the previous models of Gabrovšek et al. 2018, which exploited P1 as a unique control point. We tried to assess the error on the water budget of the system by considering the P1 to be representative of the level in the whole last chamber of the Škocjan caves (Martelova) and SK1 to be a proxy of the level in Sumeča and Hanke channels. We assumed the RMS difference between observed and modelled gravity signals (40 nm/s²) as a proxy of the error on the modelled water level in Sumeča Jama. Considering the geometry of our model and the position of the gravimeter, a 40 nm/s² gravity variation translates to a water variation of about 10 m, which propagates to a water volume uncertainty of $1.3 \cdot 10^5$ m³. For the Hanke channel, since it has a similar volume to Sumeča, we can assume the same uncertainty in water volume budget estimation.

We estimated a water volume uncertainty of $2 \cdot 10^5$ m³ for Martelova, considering the RMS difference between the modelled and observed water level (8 m) as an estimate on the modelled water level error. Consequently, the total volume budget error for the entire Škocjan caves system, in a worst case scenario, could sum up maximally to +/- $4.6 \cdot 10^5$ m³.

Another possible source of uncertainty comes from the role of sediment transport which is neglected in our modelling approach but that can be an important flux mass source as testified by Mouyen et al. (2020). The sediment mass affects the volume estimation by increasing the average density of the fluid and consequently reducing the water volume needed to explain the gravity anomaly. Modelling the sediment transport is beyond the scopes of this study and requires further observations on the deposited sediments to calibrate a hydrodynamic-sedimentological model; all these data up to now are lacking for the Reka catchment basin. In any case we tried to give at least an estimate of what could be the impact of the suspended sediments in the water mass budget estimate. Petkovšek and Mikoš (2003) measured the suspended sediments load during several floods occurring in another river in Slovenia (Dragonja) in a similar flysch context. The river, similar to Reka, has a torrential regime with sudden variations in the discharge and sequent flood pulses propagating through the river channel; the authors measured the suspended sediment during six flood events with different peak discharge. The largest event had peak discharge of over 20 m³/s while the other relevant events range from 2 m³/s to almost 5 m³/s. During the 20 m³/s event the average concentration of sediments was about 15 kg/m³, a value that drops to about 3 kg/m³ for the 5 m³/s event. The relation between peak discharge and sediment is not expected to be linear especially for very large events, however a linear approximation is the best we can do up to now. Therefore, we fit a relation between discharge and maximal sediment concentration for the Dragonja river and calculate the suspended load corresponding to a discharge value of 300 m³/s. We found that in this case the sediment load should amount up to 120 kg/m³. This translates to a volume reduction of 12% ($4.2 \cdot 10^5$ m³) at flood's peak volume, which is in any case within the estimated bounds of uncertainty.

## 5 Discussion

### 5.1 Implications of gravity measurements for the Škocjan hydrodynamics

The water accumulation in the Škocjan caves exceeded $3.5 \cdot 10^6$ m$^3$ in the 2019 event, which was stored for up to 12 h in the cave system. The temporary accumulation of water volume in the three chambers of the caves generates a gravity transient with an amplitude of almost 400 nm/s$^2$ and explains up to 85% of the observed gravity signal. The remainder portion of the signal is mostly due to Reka water mass variations occurring in the two dolines located before the entrance of the cave system. Our gravimetrically constrained hydraulic model confirms the dominant role of the allogenic contribution in Škocjan. For large flood events, as the one observed, the meteoric water masses seem to be mostly channelized and conveyed to the underground canyon. The autogenic recharge plays a secondary role, and, unlike the Rochefort case (Watlet et al., 2020) there is no need to hypothesize further large unknown karstic structures nearby the station able to temporarily store meteoric water during rain events.

In any case, similar to the Rochefort cave we observe the presence of a slight positive gravity anomaly (about 20 nm/s$^2$) after the flood event, which is not explained by our hydraulic-gravimetric model. This could be an effect of the delayed infiltration of the rainwaters in the diffuse and less permeable matrix of the karst aquifer. Compared to the allogenic recharge this contribution seems to be about 20 times lower.

Including the gravimetric observations allowed to obtain refined flux mass estimates, improving the previous model which was exclusively constrained by hydrologic data. With the gravity data from SK1 the monitoring network increased by a further independent observation which is informative of a different sector of the cave with respect to P1, which is clearly representative of the level in Martelova. The different sensitivity of measurements is also testified by the different duration of the transients recorded by P1 and SK1.

Our hydraulic model confirms the role of the conduits system downstream Škocjan in regulating its hydrodynamics. With respect to the original model of Gabrovšek et al. (2018), apart from improving the volume discretization of the Škocjan caves, we also had to adjust length and cross-sections of some conduits downstream the cave system in order to improve the fit of the observations. In particular, we had to elongate a bit the conduits connecting Škocjan to Kačna, but the most important change regarded the channels downstream Kačna, towards Kanjaduce, for which we needed to almost halve the cross-section available for flow. This change was indispensable in order to improve the fitting of the timing of the flood event, and in particular to reproduce the time delay between the Reka discharge peak and the peaks of gravity and water level observed in SK1 and P1. This further suggests that the response of the system depends on the structure of the drainage system even at distances exceeding 5 km from the observation target.

At the end, our hydraulic model captures the main features of the flood event; further improvements of the fit can be achieved by implementing an inversion procedure, optimizing the channel geometries in the less known part of the system. In any case given the large amounts of parameters of the hydraulic model and their trade-offs, some more constraints should be placed at

least for the Kačna system which, given its easier accessibility, can be realistically better characterized in hydrodynamic terms with respect to the portions of the system downstream.

**5.2 Detectability of water storage units in karst through gravimetry**

The occurrence of large voids in the epi-phreatic zone connected by networks of conduits - able to store large amounts of water during the recharge process - is not a unique characteristic of the Škocjan and Kras/Carso areas. It is a common feature of many karstic regions all over the world (e.g. the river Lomme in Belgium (Watlet et al., 2020)). Whether such voids can be detected by transient gravity signals, depends on their volume, the extent of groundwater fluctuation, and their distance from

the surface. The hydrodynamics of several other caves in the Classical Karst can be effectively remotely monitored with a similar approach.

Figure 6a plots the expected gravity signals, approximated by a spherically water-filled cavity (density contrast 1000 kg/m$^3$), as a function of the stored water volume and of the centroid depth of the cave system; a sketch of the geometries for the caves is shown in Figure 6c. We also report the simulated gravity variations due to water accumulation in three more caves in the

Classical Karst area: Kačna Cave, the Trebiciano Abyss and the Lindner Cave.

The Trebiciano Abyss is a cave of over 330 m depth, about 10 km NW from Škocjan Caves. A system of small vertical shafts connects the surface to a large chamber with a volume > 300,000 m$^3$ located 12 m above the sea level. Two lakes at the bottom of the chamber are hydraulically connected to the Reka/Timavo flow. During extreme floods, the water level can rise over 100 m, completely flooding the chamber. The Lindner Cave is located in the final sector of the Reka-Timavo path and it is

constituted by a large gallery located few meters above sea level connected to a large sub-superficial chamber by a vertical shaft. The storage unit is smaller compared to both Trebiciano and Kačna and from the internal surveys the capacity should be around 80,000 - 100,000 m$^3$. During flood events Kačna Cave generates a clearly detectable gravity signal at surface (> 500 nm/s$^2$), while in Trebiciano the gravitational signal triggered by a large flood event would be lower (200 nm/s$^2$) but still clearly detectable by a common spring gravimeter. The gravity signal associated to the Lindner cave is about 100 nm/s$^2$. We remark

that in these simulations the implicit assumption is that the gravimeter is located exactly above the cave's centroid, since as the gravimeter is moved away from the cavity axis, the signal amplitude decays.

Figure 6b finally shows gravity profiles for the different caves as a gravimeter is moved away from the cavity axis. Dashed grey and purple lines mark the expected noise thresholds for a spring gravimeter (1 nm/s$^2$, according to our noise estimate in the diurnal spectral band) and for a SG, respectively. For the SG we assume the noise threshold to be lower by 1 order of

magnitude, according to the SG noise spectra reported in Figure 3a and, hence, at the level of 0.1 nm/s$^2$. We see that hydrological signals are detectable by common spring gravimeters also some several hundred meters away from the water filled cavity; for a SG the radius of detectability is easily increased by a factor 3.

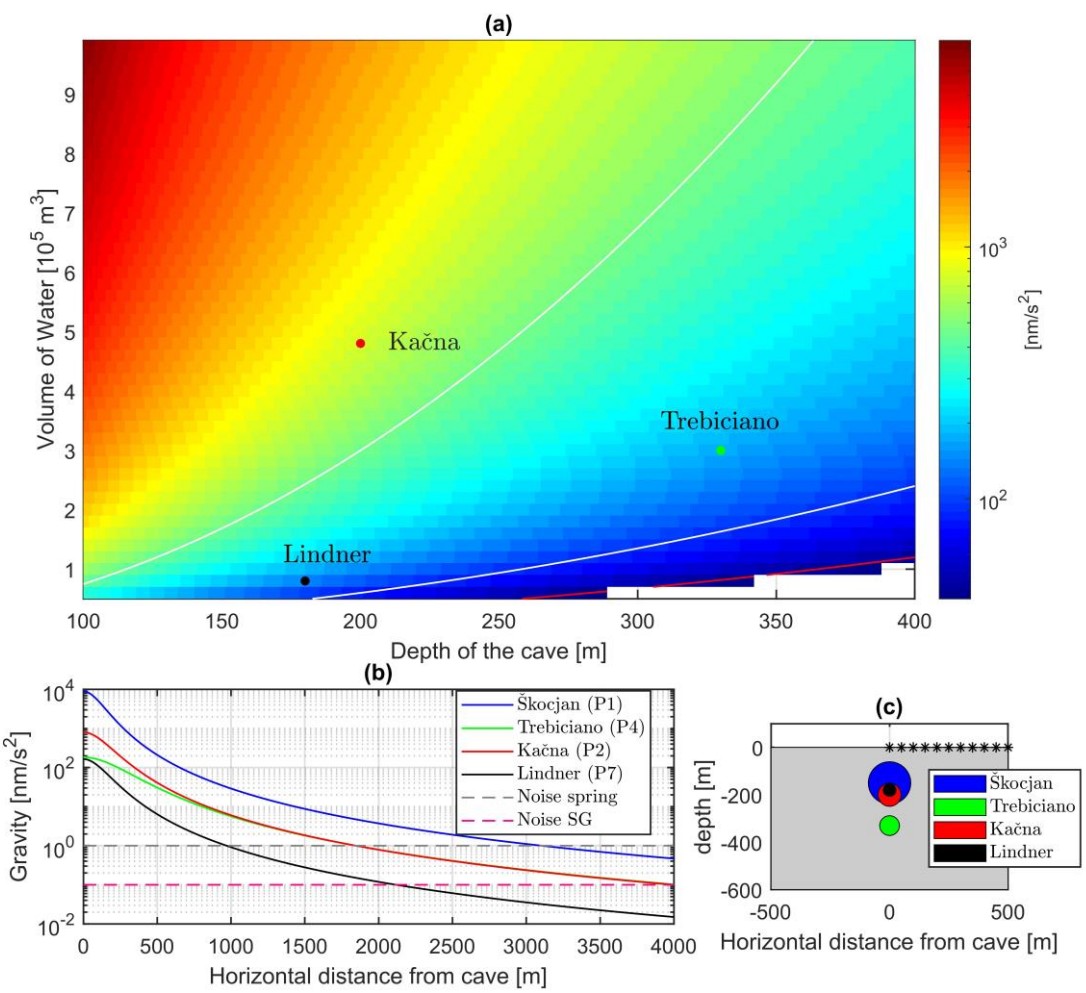

**Figure 6 Detectability of gravity signals related to hydrology in karstic areas. a) Gravity signals simulated for various water volumes stored and depths of the caves. Red line: contour line of 50 nm/s$^2$; white lines: 100 and 500 nm/s$^2$ iso-anomalies. b) Gravity anomalies for 4 different cavity volumes filled with water at different depths as a function of the horizontal distance from the cavity centre. Noise levels for a common spring gravimeter and a SG are reported with the dashed lines. c) Vertical cross-section of the spherical cavities used for approximating the hydrologic gravity effects: black stars report the surface locations where gravity anomalies have been simulated.**

In addition to the noise level of the instrument employed for the monitoring, the superposition of other geophysical phenomena in the same spectral band poses further limits to the detectability of the hydrologic transients. Apart the atmospheric effects, which in principle can be corrected with precisions up to 1 nm/s$^2$ (Watlet et al., 2020), a serious limiting factor for coastal karst

areas is due to NTOL, which can be hardly well predicted by global models. For instance, in the Classical Karst, the NTOL

from the nearby Adriatic Sea is frequently correlated with the passage of weather systems responsible of the hydrologic signals

as well. For SK1 the NTOL, at least for the investigated flood has amplitudes below 10 nm/s$^2$ and hence clearly is a secondary

effect compared to the flood's amplitude signal. However, for other cavities closer to the coast, the NTOL becomes

significantly higher and in principle, it could cause transients with amplitudes similar as those from the hydrology. An increase

of +1 m of the North Adriatic Sea level (frequently observed during low pressure fronts and sustained winds from south) can

theoretically cause gravity variations exceeding 20 nm/s$^2$ in the vicinity of the coast, which is about 1/5 of the signal of the

Lindner Cave. In the Classical Karst case an empirical model of the NTOL constrained by tide gauge observations is required,

given the limited temporal and spatial resolutions of the NTOL global models. Obviously, an empirical model has to deal with

the uncertainties due to the interpolation, which at the end result in a higher detection threshold for the hydrologic signals.

The detected signal then can be modelled in order to get quantitative estimates of the water fluxes occurring in a target cave

of interest during a flood event. The modelling procedure needs to be constrained by a hydraulic model that requires

information on the geometry of the cave system. With the Škocjan case, we demonstrated that reliable results are obtained by

approximating the cave system volume through a series of interconnected conduits with elementary cross-section shapes. For

less known areas, adequate estimates of the subsurface volume distribution could be derived from geophysical prospections

such as gravity static surveys, which offer volume estimations with sufficient precision (Braitenberg et al., 2016).

Obviously, it is important to have a picture of the karstic structures nearby the cave system also for the interpretation of the

observed transient, since part of it could be caused by other sources than the monitoring target.

**6 Conclusion**

Karstic areas are characterized by the presence of large voids, which are able to temporarily store significant water volumes

during meteorological events. Usually, a small portion of the system is known by direct hydrological observations, while a

large part is just inferred. The gravity data in Škocjan allowed to obtain a refined estimate of the water flux in the cave during

an extreme flood event of the Reka River demonstrating the added value of integrating hydraulic models and gravity and

hydrologic observations. The recorded transient, when interpreted with hydraulic models, unveils rather complex

hydrodynamic mechanisms, such as back-flooding effects, which are responsible for the accumulation of large water volume

in the system. Additionally, the gravity is also informative of other processes occurring in the vadose zone as the slow seepage

of infiltrated waters within the massif that seems to cause the slight positive anomaly after the flood event.

Through the Škocjan study case we showed that a limited number of conduits and restrictions controls the hydraulics of such

a system and reliable mass fluxes can be obtained by integrating such hydraulic models. This is an important outcome since it

suggests that indirect methods such as gravimetry could be employed in combination with these models to study parts of the

karstic aquifers where only few or weak constraints on hydrology are available. For the Classical Karst several other caves

could benefit from a similar integrated approach of the study; the expected hydrologic related signals are above the noise level

of both superconducting gravimeters as well as of the cheaper spring-based gravimeters. Apart from karst aquifers in carbonates, a similar approach can be extended to monitor cavities on gypsum and evaporates, which represent a hazard in many regions worldwide.

## Data availability

The gravity dataset composed by observations and various corrections are available upon request to the authors.

## Acknowledgments

A PhD grant to author Tommaso Pivetta was provided by Regione Friuli-Venezia Giulia (Italy) through an European Social Fund 50% cofunded fellowship. Regional Code: F17101346001. The authors acknowledge the project L7-8268 (Karst research for sustainable use of Škocjan Caves as World heritage) that was financially supported by the Slovenian Research Agency. Detlef Vogel and Jan Bergmann-Barrocas are greatly acknowledged for the installation and maintenance of the station acquisition system. Finally, we appreciate the comments of the reviewers Nolwenn Lesparre and Gerhard Jentzsch that definitely helped to improve the paper in various aspects.

## Author contributions

TP, CB, FG and GG: study design and experimental set up. TP conducted the analysis and wrote the first draft of the manuscript. BM and TP performed the tidal analysis and applied non-hydrological corrections on data. TP, FG, CB, GG: analysis of hydrologic related gravity transient and modelling. All authors contributed to the article and discussion on the data.

## Competing interests

The authors declare that they have no conflict of interest.

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

## Appendix A: Gravity Observations and reductions

Figure A1 shows the gravity observations and the applied reductions for a one-month time-series from 15th January to 15th February 2019. The time span includes the flood event that generated the huge transient gravity signal, which is already evident
in the raw observations (blue line in A1a). In the same plot, the Earth and ocean tides prediction obtained from the tidal analysis is shown in red; the tidal parameters were determined by the analysis of 2-years observations through ETERNA- ET34-X-V73 software (Schüller, 2015). The original 1 Hz data were down-sampled to 1 hour sample rate and cleaned from seismic noise, steps, and small gaps exploiting the remove-restore technique (Hinderer et al., 2007). We then proceeded to the tidal analysis for estimating amplitudes and phases of 16 main tidal constituents in the diurnal, semi-diurnal, and ter-diurnal frequency bands.
The oceanic loading contribution was removed from the tidal parameters by including the FES2014 model correction, provided by the Onsala Space Observatory (http://holt.oso.chalmers.se/loading/). Long period tides were not considered in the tidal analysis since the gPhone shows a highly irregular long-term drift and our main focus is on short term hydrologic variations. The tidal analysis allowed to detect a small calibration issue of the instrument which in fact lead to a correction factor reducing the original gravity signal by about 1%. The estimated amplitude factors and phases are reported in Table A1 together with
their standard deviations. Both the original results and the results corrected for the calibration factor are shown. In the table we report also the tidal parameters corrected for the Oceanic loading effects, the theoretical $\delta$ factors calculated from the Dehant-Defraigne-Wahr non-hydrostatic, inelastic Earth Model (Dehant et al., 1999), the residual vector length (in percentage with respect to the observed amplitude) and the residual vector phase. For a detailed description of such vectors we refer to Meurers et al. (2016). We observe that the consideration of the Oceanic Loading correction improves the $\delta$ factors of several
tidal constituents, which now are very close to the theoretical ones.

The overall quality of our Local Tidal Model (LTM) is testified by the residual spectrum (yellow line in Figure A1b), i.e. the difference between the spectra of the observed data and the synthetic tidal model, which shows the strong reduction of the signal energy in the tidal frequency bands.

The residual time series for the January-February 2019 period after removing the LTM is shown in the Figure A1b by the blue
line; here we also observe the >400 nm/s$^2$ gravity anomaly peak associated with the flood event in Škocjan and other smaller transients that are highly correlated with atmospheric pressure variations.

The gravity changes due to atmospheric pressure variations (Figure A1c) have been calculated and removed by integrating an empirical local admittance with the global ATMACS model (Klügel and Wziontek, 2009). We followed the approach described in Karbon et al. (2014) in which the authors firstly interpolate the ATMACS loading and Newtonian components from the original 3h sample rate to 1h temporal resolution. Then the pressure difference between the interpolated ATMACS model and the observed pressure by the co-located barometer in SK1 is corrected through the admittance estimated by the tidal analysis. The empirical admittance, an estimation in the diurnal and semi-diurnal frequency bands for SK1, is equal to -3.39 +/- 0.05 $nm/s^2$/hPa which is close to the theoretical value (Hinderer et al., 2007) and, thus, proves that the instrument is well sealed. As evident from Figure A1d, the gravity effect due to atmospheric pressure changes shows smooth variations of 50-60 $nm/s^2$ with periods of about 4-5 days; however sharper changes occur during important meteorological events when the variations can exceed 80 $nm/s^2$ over periods of 1-2 days. The effects of the atmospheric pressure reduction to the SK1 gravity data can be seen in Figure A1c by comparing the residuals before (blue line) and after (red line) applying the corrections. The improvement is testified by the reduction of the signal energy from the original 60 $nm/s^2$ to 54 $nm/s^2$ after the reduction.

In the same Figure, we show the gravity residuals after removing the global hydrologic contribution, which is due to the Newtonian and loading effects of the continental water masses at distances larger than 0.1° from SK1. The effect of the water masses is estimated relying on the GLDAS database (Rodell et al., 2004) that offers a worldwide model of the soil moisture variations with temporal resolution of 3 hours. The gravity effect is very small compared to the other corrections and in the time span considered in Figure A1c, it has a standard deviation of 1.5 $nm/s^2$ and maximal excursion of 5 $nm/s^2$.

The last Figure A1e shows an estimation of the local water storage near the station SK1 during a rain event. We calculated the gravity response to a 1 mm rain event, using a high resolution topographic model, as a function of the radial distance from SK1 (Meurers et al., 2007). The curves in Figure A1e show the admittances as a function of the radius of the areas considered. Two scenarios are examined: one considering the "umbrella effect" which assumes the immediate surrounding of the meter to be shielded and hence not able to temporarily store water masses, the other assuming the umbrella effect to be negligible. The first simulation is the most realistic in our case since the instrument is located in a building (surrounded by parking lots and several other infrastructures) that shields the ground; this scenario predicts for a 7 mm rain event (peak recorded in Škocjan during the flood event) an effect below 1 $nm/s^2$.

We finally evaluated the non-tidal oceanic contribution, since the Adriatic Sea is less than 30 km away. We relied on the observed sea level data at 5 different harbours along the Adriatic coast (Koper, Trieste, Monfalcone, Venezia, and Ancona) and we calculated the non-tidal component by removing the tidal signal estimated from tidal analysis. From each time-step of these time-series we interpolate the sea level heights on the whole Adriatic basin on a regular grid of 0.025°. A tesseroid discretization was employed to compute the Newtonian effects while the deformation effect was taken into account by convolving the mass loads at each time-step with the elastic Green functions. For the whole time span 15th January -15th February 2019 we observed coherent variations in the sea level at all the stations: during the flood event there was a non-tidal transient of amplitude 0.6 m that lasted for about 12 hours, recorded at Koper, Trieste, and Monfalcone stations. We computed

the gravity contribution of this signal to be less than 10 nm/s$^2$ at SK1. After applying all the reductions, the final standard deviations of the residuals amounts to 20 nm/s$^2$ for the period considered in Figure A1a.

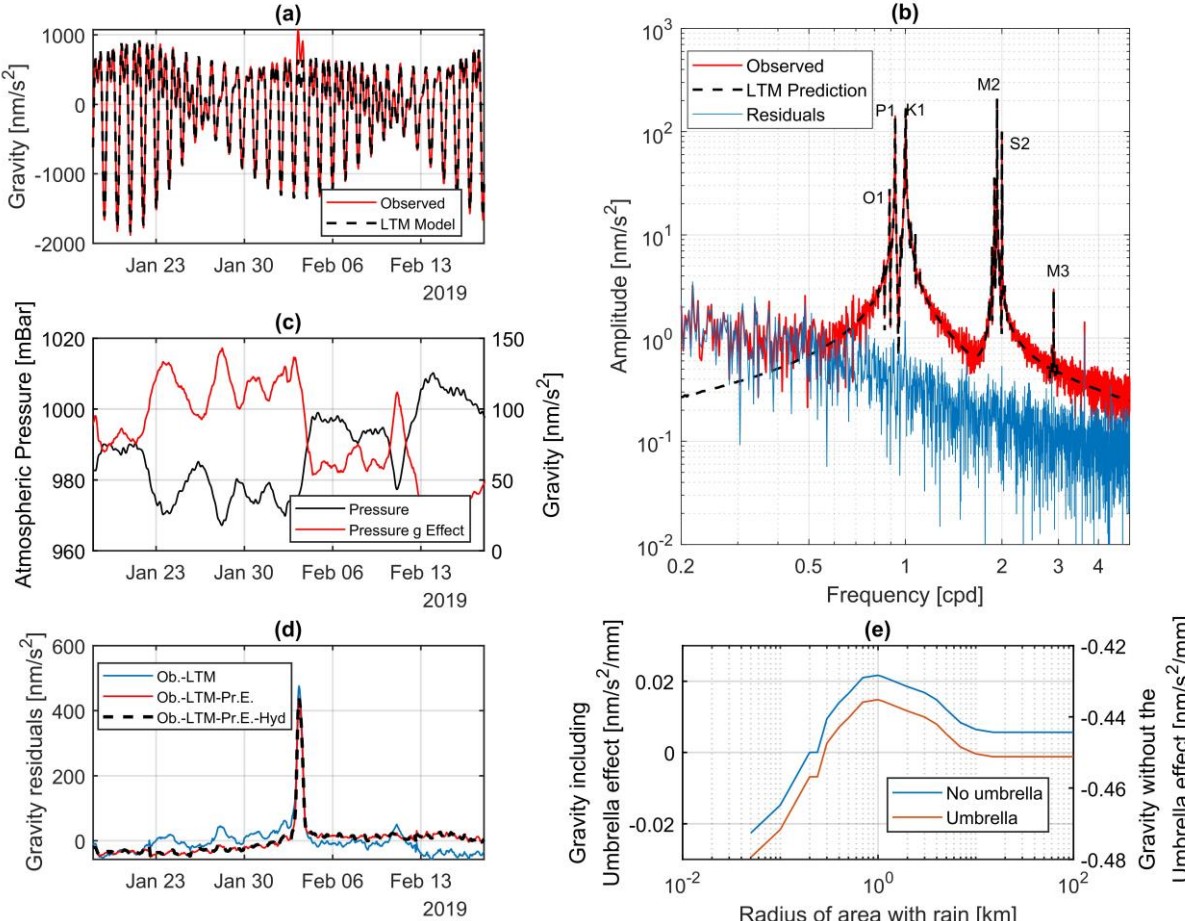

**Figure A1 Gravity observations and reductions of SK1. a) observed signal in red and Local Tidal model (LTM)**
**prediction in black. b) spectra of the observed data (red), predicted tides (dashed black), and residuals (blue). O1, P1,**
**K1, M2, S2, M3 are the main tidal constituents observed in the data (see also Table A1). c) atmospheric pressure time-**
**series and related gravity effect at SK1. d) gravity residuals after successively removing the LTM (blue), the**
**atmospheric pressure variations (red), and the global hydrologic contribution (black dashed). e) gravity response to 1**
**mm rain event at SK1 as a function of the radius of the area where it is supposed to rain; red curve: umbrella effect**
**considered; blue: no umbrella effect.**

| Wave Group | freq.[°/h] | Amplitude Theo. [nm/s²] | Amplitude Obs. [nm/s²] | $\delta_o$ | $\delta_o$ RMS | $\kappa_o$ [°] | $\kappa_o$ RMS [°] | $\delta_o$-OL. | $\kappa_o$-OL [°] | $\delta$ DDW-NHi | $\delta_o$ SF corr.-OL | $\chi$: residual vector length % | residual vector phase [°] |
|---|---|---|---|---|---|---|---|---|---|---|---|---|---|
| **Diurnal Band** | | | | | | | | | | | | | |
| $Q_1$ | **13.3987** | **59.472** | **69.066** | **1.1625** | **0.0036** | **0.087** | **0.168** | **1.1682** | **0.158** | **1.15428** | **1.15360** | **12.5** | **29.5** |
| $O_1$ | **13.943** | **310.616** | **361.633** | **1.1651** | **0.0007** | **0.163** | **0.033** | **1.1698** | **0.011** | **1.15427** | **1.15426** | **2.3** | **1.7** |
| $M_1$ | 14.49669 | 24.417 | 28.994 | 1.17597 | 0.00831 | 0.036 | 0.394 | | - | - | 1.16102 | - | - |
| $P_1$ | **14.9589** | **144.506** | **169.213** | **1.1661** | **0.0014** | **0.044** | **0.023** | **1.1676** | **-0.302** | **1.14912** | **1.15336** | **58.8** | **93.2** |
| $S_1$ | 15 | 3.414 | 6.3253 | 1.46956 | 0.08654 | -36.395 | 3.293 | - | - | - | 1.41905 | - | - |
| $K_1$ | **15.0411** | **436.679** | **503.759** | **1.1527** | **0.0005** | **0.396** | **0.024** | **1.1538** | **0.044** | **1.13470** | **1.13880** | **18.8** | **29.9** |
| $\psi_1$ | 15.08214 | 3.415 | 3.039 | 1.0157 | 0.06185 | 9.027 | 3.186 | - | - | - | 1.05071 | - | - |
| $\varphi_1$ | 15.12321 | 6.217 | 8.254 | 1.28922 | 0.03275 | 2.159 | 1.394 | - | - | - | 1.26498 | - | - |
| $J_1$ | 15.58544 | 24.425 | 28.458 | 1.16206 | 0.00854 | 0.906 | 0.406 | - | - | - | 1.14007 | - | - |
| $OO_1$ | 16.1391 | 13.357 | 15.762 | 1.19573 | 0.0163 | 1.223 | 0.763 | - | - | - | 1.17576 | - | - |
| **Semi-Diurnal Band** | | | | | | | | | | | | | |
| $2N_2$ | 27.96821 | 11.226 | 13.21 | 1.17766 | 0.00541 | 1.308 | 0.294 | - | - | - | 1.16198 | - | - |
| $N_2$ | **28.4397** | **70.292** | **83.596** | **1.19** | **0.0011** | **1.417** | **0.058** | **1.1763** | **-0.078** | **1.16194** | **1.16069** | **5.8** | **0.9** |
| $M_2$ | **28.9841** | **367.124** | **439.658** | **1.1982** | **0.0002** | **1.012** | **0.012** | **1.1783** | **-0.009** | **1.16194** | **1.16282** | **1.7** | **1.5** |
| $L_2$ | 29.52848 | 10.378 | 12.503 | 1.19185 | 0.01078 | 1.785 | 0.585 | - | - | - | 1.17395 | - | - |
| $S_2$ | **30** | **170.79** | **204.29** | **1.1962** | **0.0005** | **0.371** | **0.329** | **1.1766** | **0.113** | **1.16194** | **1.16278** | **7.1** | **5.5** |
| $K_2$ | **30.0821** | **46.398** | **55.272** | **1.194** | **0.0019** | **0.231** | **0.189** | **1.1739** | **-0.073** | **1.16194** | **1.15912** | **14.9** | **2.3** |
| **Ter-Diurnal Band** | | | | | | | | | | | | | |
| $M_3$ | 43.47616 | 5.044 | 5.528 | 1.0995 | 0.008 | 0.470 | 0.417 | | | | 1.08118 | | |
| | Atmospheric pressure admittance | | | -3.39 +/- 0.05 nm/s²/hPa | | | | | | | | | |

**Table A1 Tidal analysis results. δₒ= δ observed; κₒ=phase observed; δₒ-OL= δ observed corrected for oceanic loading (OL); κₒ -OL= phase corrected for OL; δ DDW-NIh= theoretical response; δₒ SF corr.-OL= δ observed corrected for Scale Factor and OL; χ: residual vector length. For definition of χ and phase residuals please refer to Meurers et al. (2016). Rows in bold = FES2014b correction available.**

## Appendix B: Model of the flood event

In the manuscript we present and discuss the results of a simple hydraulic model which indeed was sufficient to get a good fit of the observed data and also to get an estimate of the water volume balance of the caves during a flood event. As already hinted, the Škocjan study case suggests that a limited number of conduits and restrictions mainly controls the hydraulics of such a system. However, to strengthen and prove this argument we calculated the hydraulic and gravimetric response of a model that includes a more realistic definition of the channel geometries. The model relies on several topographic measurements that mapped both the areas surrounding the Škocjan caves and the main internal morphological structures: the measurements were performed employing classical topographic instruments such as total station, GNSS and a laser profiler. Figure B1a gives an overview of the available data: the red lines show the location of 25 cross-sections while the green dots report the location of various levelling points. The whole survey (cross sections and levelling points) is constituted by a point cloud of over 3000 topographic points which mapped the internal morphologies of the cave and the topography of the dolines which host the Reka river before entering the cave system.

The hydraulic response to the flood event was calculated using the SWMM routines: in this case we discretized the Škocjan caves canyon with 11 conduits (thick red lines in Figure B1a) of different cross section shapes, each of them constrained by the internal topographic survey. The cross-section areas for the 11 sections are reported in Figure 1b while Figure B2 shows six exemplary cross sections. For the external area, the channel geometry was derived from 3 topographic profiles taken from the DEM model (ARSO). The data from the hydrograph at Cerkvenikov Mlin was set as input flow at the first node of the model that is located 5 km away from Škocjan. We simulated the water heights in the different conduits from the 1st February to the 5th February with hourly resolution. The comparison between observed and modelled water level in Martelova is shown in Figure B3d and shows a similar fit as the simple model.

The calculation of the gravimetric effect requires the definition of two surfaces for determining the geometry of the channels: one for the bottom where the river is flowing, and the other for the ceiling of the cave. We obtained these two surfaces by firstly dividing the point cloud between ceiling and bottom exploiting a surface that locally follows the median plane of the cave. A similar approach was used by Pivetta and Braitenberg (2015) to process the point cloud of laser-scan data from the Grotta Gigante. Then we interpolated the scattered points separately into two regular grids with spatial resolution of 2 m × 2 m. The two surfaces are plotted in Figure B3a and B3b; the colour code is proportional to the height of the surface from sea level. The ceiling appears to be smooth since it is constrained by fewer observations than the bottom. The total volume of the caves from this model is about $5 \cdot 10^6$ m$^3$. For the areas external to the caves, the bottom surface of the riverbed is obtained by integrating the topography derived from the laser-scan survey (ARSO).

The gravity effects at SK1 were calculated using a prism discretization of the whole water mass of the model at each time step. The prism model is constrained by the water level simulated in each conduit, and by the cave's bottom and ceiling surfaces. In this case, with respect to the simpler model presented in the manuscript, the flooded area in a specific section of the cave

could change as a function of the water level in the conduit. An example is given in Figure B3b where the black and red outlines bound the flooded areas at two different moments during the flooding: the first is during the rising phase when the

785 water level in P1 is 50m while the second is at the flood's peak. The modelled gravity time-series are shown together with the observations in Figure B3e: the red curve illustrates the gravity signal caused by considering only the water masses inside the cave while the blue line reports the effect including the masses stored outside. By 'outside' we mean the water masses contained inside the blue dashed outline in Figure B1a. Our model reproduces fairly well the shape of the observed gravity transient while it underestimates a bit the magnitude of the event; this is probably attributable to the simplifications of the hydraulic

model in the external areas near the two dolines (MD and VD). The final RMS error for the gravity time-series considering all the masses amounts to 30 nm/s$^2$.

The last Figure B3e shows, with the same color code as in B3d, the mass flux time series: we see that the estimated water volumes stored in the cave during the peak flood amount to about $3.5 \cdot 10^6$ m$^3$ being in agreement with the estimation discussed in the manuscript.

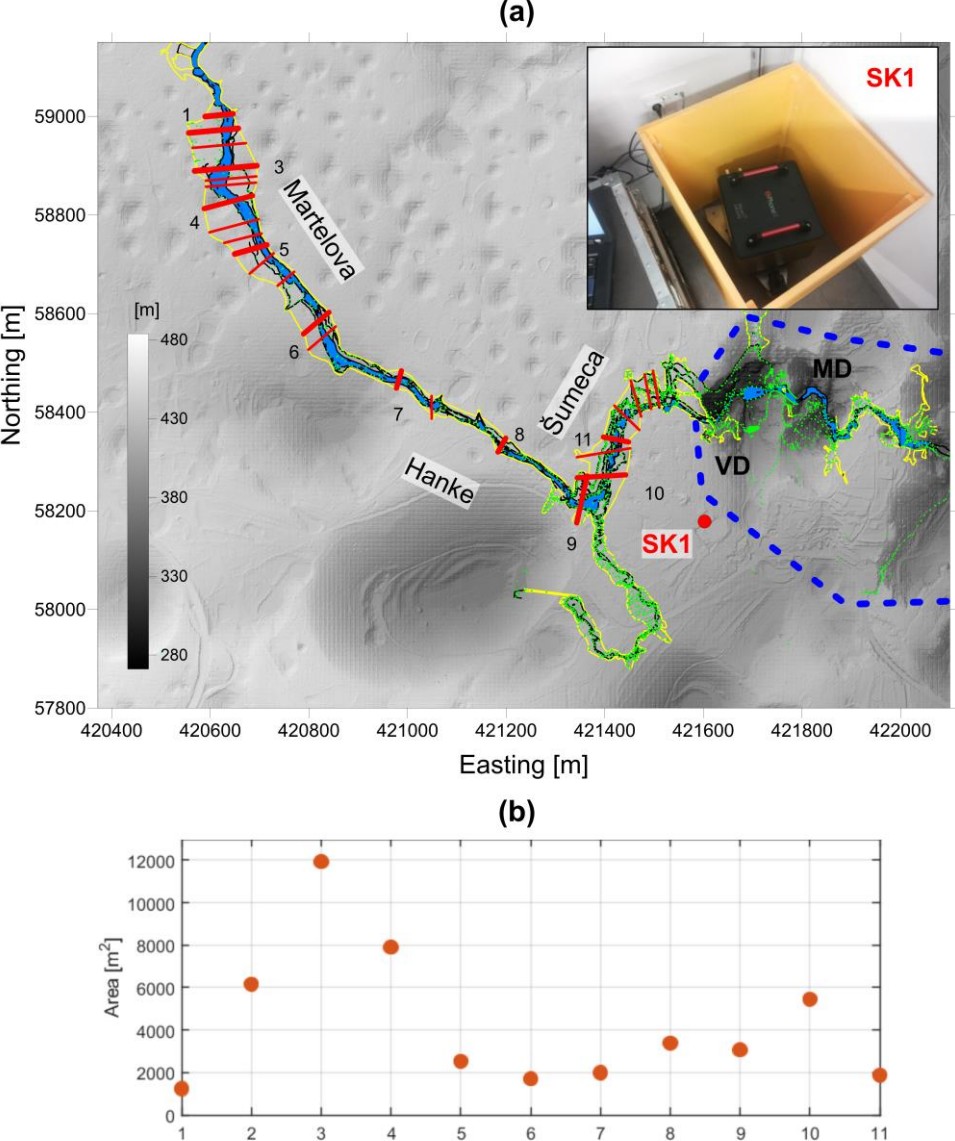

**Figure B1 Topographic data available in the Škocjan area. a) DEM of the Škocjan area (shown through a gray-scale colour code) with the outline of the cave (yellow), the position of the topographic points (green dots) and the location of the vertical cross-sections of the channels (red lines). Thick red lines show the locations of the 11 cross sections used for constraining our hydraulic model. Blue dashed line bounds the area of the topography included in our model. VD=Velika Dolina; MD= Mala Dolina are the two coalescent dolines located at the beginning of the cave system. SK1 position shown with the red dot. b) surface areas of the various cross sections calculated from the internal topographic survey.**

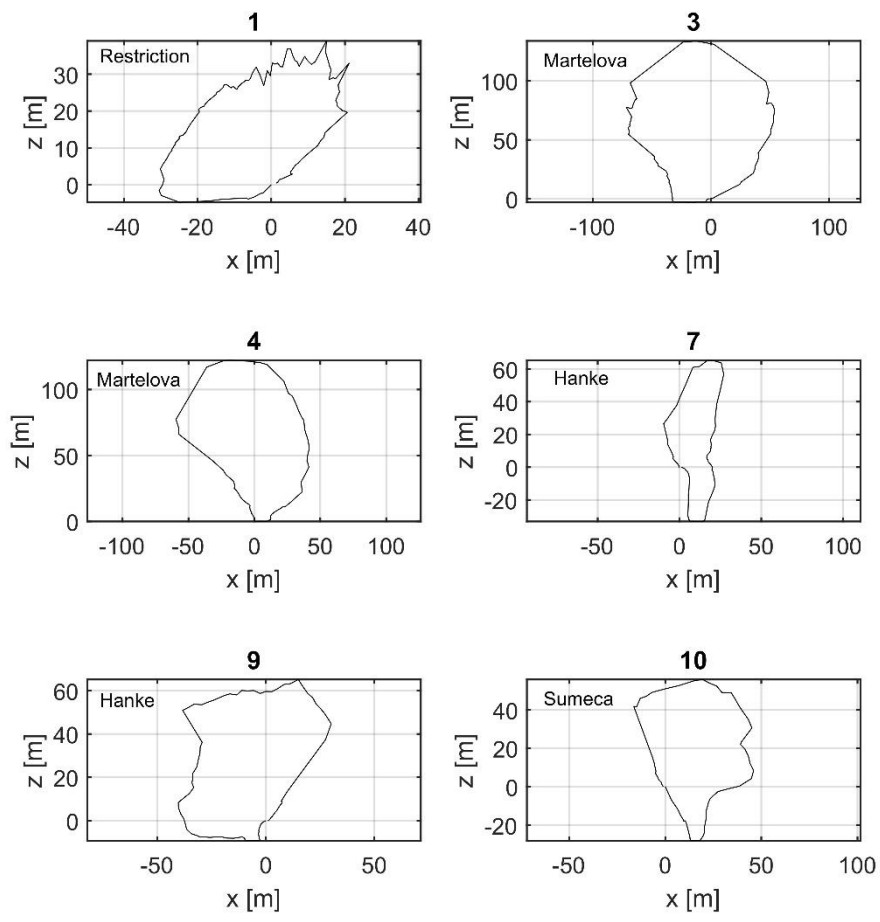

**Figure B2 Examples of the cross section of the cave system; location labelled in Figure B1a.**

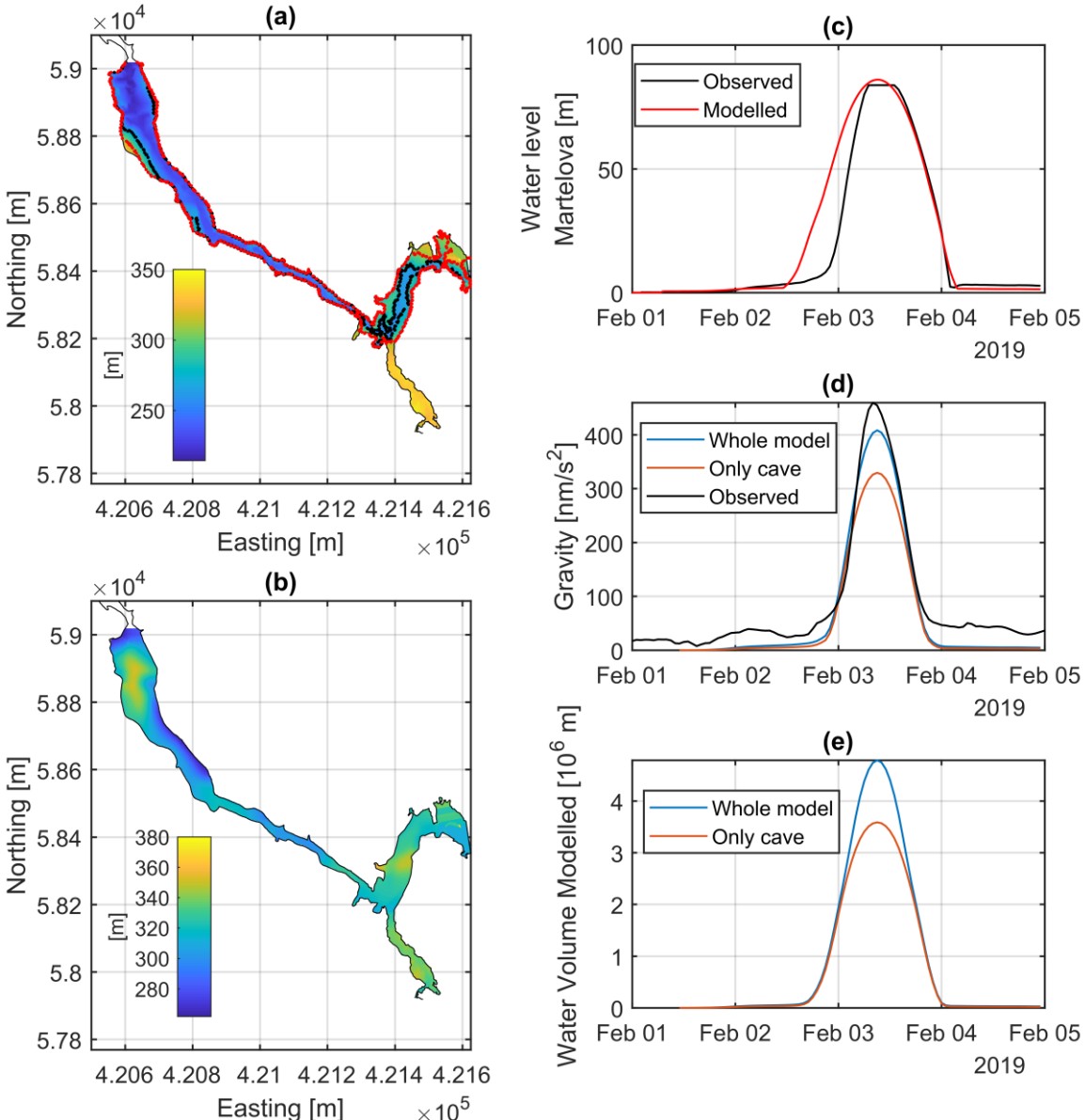

**Figure B3 Modelling the flood event. a) height above sea level of the bottom of the channel where the Reka flows; the red and black outlines show the flooded area when the water level in P1 is respectively 86 m and 50 m. b) height above sea level of the ceiling of the cave. c), d) and e) results of the hydraulic model calculated taking into account the new 3D model of the cave. c) observations (black) and modelling (red) of the water level time-series in P1 d) black: observed gravity residual; blue: modelled gravity transient for the whole model which includes the area outlined in B1a); red: gravity effect of the solely masses inside the cave. e) water volume fluxes during the flood; the color code as in e).**
