# Peer review of "Gravity as a tool to improve the hydrologic mass budget in karstic areas"

_Hydrology and Earth System Sciences, 2021_

## Author Comment (AC1)

➢ Reply:

➢ We thank reviewer Dr. Nolwenn Lesparre for the general appreciation of the work and for the interesting comments and suggestions, which we all incorporated into the manuscript. Here we give a detailed response to all the comments. The original comments are given in black color, our responses in red color.

Reviewer comment by Dr. Nolwenn Lesparre (https://doi.org/10.5194/hess-2021-294-RC1):

The paper "Gravity as a tool to improve the hydrologic mass budget in karstic areas" presents an original data set of continuous gravimetric measurements acquired above karstic caves during a flood event. The authors developed a 3D hydraulic model to be able to estimate the water level monitored in the karstic conduit as well as the gravity data set. From the model they infer the water volume flowing through the karstic network during the flood event. They show that the karstic cave above which they performed measurements is particularly suitable for applying such a methodology thanks to its location relatively close to the surface and its large dimension. But they also show from synthetic tests that other caves smaller or deeper could be monitored with continuous gravimetric measurements.

The data set is particularly interesting and the model conception to reproduce the data set is pertinent. The figures are clear, the references are appropriate, but the text could gain clarity. Also the originality of the experiment should be better empasized. Find below some comments, hoping they could help you improving such weaknesses.

➢ We thank for these very encouraging comments and hope that our detailed changes will improve the clarity of the text.

Abstract: "We demonstrate how the inclusion of gravity observations improves water mass budget estimates"

→ In the paper you show (Fig. 2) how the water volume estimated differs if computed from the gravity & hydraulic model with respect to the hydraulic only model. However you do not show how the hydraulic only model fits the data (at least the water level of Martelova). Is the fit of this data set degraded when you fit also the gravity measurements?

➢ Thanks, that's a good point. We think also the description of what "only hydraulic" model means should be improved. By "hydraulic only" we mean the model of Gabrovšek et al. (2018) model which was constrained only by hydrologic observations. We specified it more explicitly in the text. Now we have added both the hydraulic and gravity fits using the original model by Gabrovšek et al. (2018). As you can see, the model underestimates both the water level in P1 and gravity. We also add a comparison between the two models in plan view and cross-section. The Gabrovšek et al. (2018) model suffers from being too rough in approximating the geometry of the cave system.

[Figure]

*Figure 2 Observed and modelled time series of the flood event in 02/2019 – red curves: new model considering gravity and hydraulic data, blue curves: former model only considering hydraulic data (Gabrovšek et al., 2018). a) Hydrological data: rain (blue bars) recorded in Škocjan and discharge (black line) at Cerkvenikov mlin. b) Observed and modelled gravity signals (black and red/blue) at SK1; t1, t2 and t3 indicate three different phases of the flood event (rising, peak, falling). c) Stage time-series recorded and modelled at Martelova/P1 (black and red/blue). d) Water volume accumulation during the event in the whole Škocjan system according to the new model (red) and based on Gabrovšek et al. (2018; blue line).*

l. 34: To assess the structure of karst aquifers, different geophysical and hydrological techniques are used, each of them being applicable to a specific situation

→ This sentence is very general and does not render the advantages and drawbacks of the different methods that could be used to study karst structures. A short description of the classical methods with references to the main contributions (such as Chalikakis, K., Plagnes, V., Guerin, R., Valois, R., & Bosch, F. P. (2011). Contribution of geophysical methods to karst-system exploration: an overview. Hydrogeology Journal, 19(6), 1169-1180.) should be added. This will help better highligthting the interest of your work and its contribution to the previous researches on karst structure studies.

➢ Ok thanks. This and the following 5 comments have been addressed by reformulating the introduction. We agree that the introduction needs a more comprehensive review of the state of the art regaring geophysics and modelling of the karst hydrodynamics. Moreover, we more explicitly defined the expected contribution and working hypothesis. So the introduction is restructured as follows:

*Karst areas on carbonates and evaporates occupy about 15 % of the ice-free continents. About one fourth of the world's and about one third of Europe's population is supplied by water from karst aquifers, where most of the water is drained through networks of solution conduits, which evolve along sedimentary or tectonic discontinuities (Ford and Williams, 2007). The evolution of conduit systems is controlled by complex mechanisms, which make the position of drainage pathways hard to predict. The evolution of karst aquifers is driven toward equilibrium, where conduit systems effectively drain all of the available recharge. However, in active tectonic environments, the evolution is being continuously stirred by changes of boundary conditions and structure (Gabrovšek et al., 2014). This makes complex geometries of networks, with high variations of conduit cross-sections and terminations of conduits by breakdowns or fault planes. Such systems are being permanently out of equilibrium, and exhibit large water level variations during flood events. Large voids with volumes in the order of $10^6$ $m^3$ are common features in such settings. The positions of solution conduits and voids in karst aquifers are largely unknown, except for the parts accessible to direct human exploration. To assess the structure of karst aquifers and the response to the recharge process, different geophysical and hydrological techniques are used, each of them being applicable to a specific situation.*

*Where the caves are directly accessible, classical hydrological instrumentation as a diver for pressure, temperature and electrical conductivity measurements may be deployed. This instrumentation is indispensable to verify hydraulic connection between cave systems during a rain event and determine the travel time of the water masses or pollutants. Such data together with meteorological observations are fundamental to constrain hydrologic and hydraulic models.*

*The identification of underground water paths can also be approached employing tracer tests and geochemical analysis of the waters (Petrič and Kogovšek, 2016; Zini et al., 2014; Goldscheider and Drew, 2014). With such techniques, hints on the water origin are obtainable; however, also these techniques require a direct access to the vadose zone, for deploying the instrumentation along the underground water paths.*

*Geophysical methods are complementary tools to these more classical hydrological prospections (Chalikakis et al., 2011): alternation of rock and voids in combination with water level variations during the recharge process cause in fact variations of the physical parameters, such as the velocity of seismic waves, change in the 
[revised manuscript text omitted]

*Deville, S., Jacob, T., Chéry, J., and Champollion, C.: On the impact of topography and building mask on time varying gravity due to local hydrology, Geophys. J. Int., 192, 82–93, https://doi.org/10.1093/gji/ggs007, 2013.*

*Fores, B., Champollion, C., Mainsant, G., Albaric, J., and Fort, A.: Monitoring Saturation Changes with Ambient Seismic Noise and Gravimetry in a Karst Environment, Vadose Zone J., 17, 170163, https://doi.org/10.2136/vzj2017.09.0163, 2018.*

*Goldscheider, N. and Drew, D.: Methods in karst hydrogeology, 280 pp., 2014.*

*Jacob, T., Bayer, R., Chery, J., Jourde, H., Moigne, N. L., Boy, J.-P., Hinderer, J., Luck, B., and Brunet, P.: Absolute gravity monitoring of water storage variation in a karst aquifer on the larzac plateau (Southern France), J. Geophys. Res. Solid Earth, 13, 2008.*

*Jacob, T., Bayer, R., Chery, J., and Moigne, N. L.: Time-lapse microgravity surveys reveal water storage heterogeneity of a karst aquifer, J. Geophys. Res. Solid Earth, 115, https://doi.org/10.1029/2009JB006616, 2011.*

*Kaufmann, G., Gabrovšek, F., and Turk, J.: Modelling flow of subterranean Pivka river in Postojnska jama, Slovenia, Acta Carsologica, 45, https://doi.org/10.3986/ac.v45i1.3059, 2016.*

*Mayaud, C., Gabrovšek, F., Blatnik, M., Kogovšek, B., Petrič, M., and Ravbar, N.: Understanding flooding in poljes: A modelling perspective, J. Hydrol., 575, 874–889, https://doi.org/10.1016/j.jhydrol.2019.04.092, 2019.*

*Petrič, M. and Kogovšek, J.: Identifying the characteristics of groundwater flow in the Classical Karst area (Slovenia/Italy) by means of tracer tests, Environ. Earth Sci., 75, 1446, https://doi.org/10.1007/s12665-016-6255-4, 2016.*

*Watlet, A., Kaufmann, O., Triantafyllou, A., Poulain, A., Chambers, J. E., Meldrum, P. I., Wilkinson, P. B., Hallet, V., Quinif, Y., Van Ruymbeke, M., and Van Camp, M.: Imaging groundwater infiltration dynamics in the karst vadose zone with long-term ERT monitoring, Hydrol. Earth Syst. Sci., 22, 1563–1592, https://doi.org/10.5194/hess-22-1563-2018, 2018.*

Paragraph from line 40: You should underline that the gravimetric temporal signal is directly sensitive to the water mass redistribution (no need of petrophysical relationships), which is very powerful when you want to interpret the variations of your signal. You could also develop the specificity of the gravimetry which is an integrative tool (versus water head which is very local), but that can still provide an information on the water content state integrated around the instrument. Discharge measurements is also integrative but it integrates information concerning the whole catchment.

➢   Ok thanks, we included this in the introduction. In particular, we added the following paragraph:

*Compared to the other geophysical techniques for monitoring underground water movements, gravimetry requires less petro-physical relations, as the change in gravity is simply related to the water density, which is frequently assumed as constant in time and space. This aspect clearly simplifies the interpretation of the observed gravity transients. Another strength of the method is related to its sensitivity to the integrated water mass around the instrument, which implies that a remote monitoring of the water storage unit is possible. This is an important aspect that allows to fill the gaps of the sparse water head observations, which depend on the accessibility of the caves in the vadose zone. Most of the gravity signal originates from the mass variations occurring just below the instrument, however the horizontal sensitivity increases as the storage unit is deeper (Van Camp et al., 2017). Being sensitive to the integrated water mass is also a key aspect to obtain a reliable mass flux balance of the system complementing the head observations, which usually are representative of a very localized portion of the system.*

You could add a paragraph on the state of the art concerning the karst hydrosystems modelling. Some of them require the geometry of the conduits, you could then develop the supply of such models compared to black boxes model that reproduce well hydrological data and do not need any geometrical information on karst networks structure.

➢   Ok thanks, we included reference to hydraulic/gravimetric modelling in the introduction that extends from the sentence "*The non-uniqueness of the inverse problem clearly affects the interpretation of the gravity observations and requires implementing hydrologic models that simulate the recharge process and the sequent groundwater time evolution.*" to … "*In our study we make use of the known 3D geometry of a cave system to interpret the gravity observations by implementing a 3D hydraulic model.*"

Paragraph from line 47: this part does not render what you expect from gravimetry to validate your model that other data type would not provide

➢ Ok thanks, we included more explicitly, what we expect to be our contribution and the working hypothesis. In particular we added this paragraph:

*Being sensitive to the integrated water mass is also a key aspect to obtain a reliable mass flux balance of the system complementing the head observations, which usually are representative of a very localized portion of the system.*

Furthermore, at the end of the introduction the following two points were added:

- *fast hydrologic variations in a cave system in the vadose zone can be effectively monitored by employing one spring-based gravimeter*
- *integration of hydraulic models, head observations and gravity helps in better assessing the water mass balance of the system*

l.52: "numerical models" → you could develop what kind of hydrological model you use.

➢ Ok thanks. In the introduction we added an extended descriptions of the hydraulic models that were employed:

*As shown by Gabrovšek et al. (2018) the water level response of such underground karstic system to the discharge variations of the Reka can be predicted using the same equations used for modelling the river hydrodynamics (i.e. Saint Venant equations)…*

l.54: "In July 2018, a continuously recording gravimeter was installed above the caves." → you could specify the duration of the continuous gravimetric measurements

➢ Ok thanks. We reformulated as follows *"In light of these considerations, in July 2018 a continuously recording spring-based gravimeter was installed in the Classical Karst area, nearby the Škocjan cave system, which is still operating. …. Up to now about 2 years of data have been analyzed and several flood events, including an extreme flood in February 2019, provided excellent gravimetric and hydrologic records that we aim to model and explain in the frame of this study."*

Fig. 1c, legend exterme → extreme

➢ Thanks, we corrected the legend entry

l. 75: Slovene /Carso → Slovene/Carso

➢ Thanks, we removed the blank space

l. 113: "The cross-section of the canyon is between 2,000 m$^2$ and 12,000 m$^2$." → could you describe how you estimate it

➢ This estimate comes from an existing internal topographic survey conducted in the past years with a laser profiler which allowed to define more than 30 cross sections along the cave system. The topographic data was also employed for constraining the hydrologic

model described in the appendix B. The cross-section areas for some selected cave sections are reported in Figure 1b, while the traces in plan view are shown in Figure 1a. Figure 2 reports six examples of the topographic cross sections.

➢ We propose to add the figures in the appendix.

[Figure]

Figure 1 a) location of sections (black lines) for which the area of the cave's cross sections is reported in (b). Yellow: Skocjan outline; thick red line: axis of the hydraulic model. b) surface areas of the various cross sections calculated from the topographic survey.

[Figure]

Figure 2: Examples of the cross section of the cave system; location shown in figure 1a.

I.124: "The long-term monitoring of groundwater" → could you indicate since when are acquired the different data (temperature/pressure in the Škocjan Caves and in the Kačna Cave, the gauging station…)

➢ Ok we included the following table which reports the main characteristics for the geophysical-hydrologic datasets acquired

| Sensor name | Measurement | Data availability intervals (month/year) | Temporal resolution |
|---|---|---|---|
| P1 | Diver for temperature and pressure | 1/2005-1/2008; 6/2008-6/2009; 1/2018-ongoing | 1 hour |
| P2 | Diver for temperature and pressure | 1/2005-1/2007; 1/2008-6/2011; 1/2013-6/2014 | 1 hour |
| SK1 | Gravity variations | 7/2018-ongoing | 1 second |
| Cerkvenikov Mlin | Discharge measurement | 1/1954-ongoing | 30 minutes |

l.141: "a gPhone gravimeter » -> could you provide the drift of your instrument and its precision

➢ Ok, we have added the in-house specifications which are: precision 10 nm/s$^2$ and drift rates < 5 10$^3$ nm/s$^2$/month. Moreover we added a more specific comment on drift, when discussing the data processing:

➢ *The drift of the instrument appears large and highly non-linear as expected from a spring based gravimeter; for the time span July 2018- August 2019 the drift followed a logarithmic-like trend which was taken into account by fitting a 5th degree polynomial curve. The first 3-months of observations confirmed the in-house specifications for the drift (around 5 10$^3$ nm/s$^2$/month); from October the drift was even lower, with values around 10$^3$ nm/s$^2$/month. Up to now, we do not have at disposal any absolute gravity measurements to check properly temporal variations of the drift and the presence of eventual long-period signals superposing on it. In any case, for the scopes of our study, exactly estimating the drift is not crucial since we are focusing on fast water mass variations that typically last for about 1-2 days when we expect the drift to be mostly linear.*

l.142: "was installed on the surface above Škocjan Caves in July 2018." → add the duration of the dataset you analyse.

➢ OK we have added the following sentence: "*Up to now we have analyzed about 2 years of data, which include 4 flood events with peak discharge < 200 m$^3$/s and an extraordinary one, which is discussed in detail in this paper and exceeded 290 m$^3$/s of peak discharge.*"

l. 170: Hydrologists might not be familiar with the term "admittance" → provide a short definition

➢ OK we have added the following sentence:

*The admittance is defined as the gravity change due to a 1hPa pressure perturbation; it is calculated by fitting a linear relation between the gravity time-series reduced for the tidal component and the pressure time-series. As detailed in the Appendix A, the value we found (-3.39 nm/s$^2$/hPa) is in good agreement with the theoretical predictions.*

Fig. 4: Could you place on the plan views the location of the cross-sections?

➢ Sorry there is maybe a misunderstanding; the vertical section follows the path of the channel system, marked in the plan view.

Could you also locate on the plan views the two dolines located before the entrance of the cave system, where water is susceptible to accumulate?

➢ OK we highlighted the location of the two main dolines.

l. 207 and Fig.3: specify what cpd means.

➢ Sorry we hadn't defined it, thanks for pointing this out. We added definition: "Cycles Per Day".

Paragraphs from l. 221 → it is not clear if you adapted the model of Gabrovšek et al. (2018) or if you built a new one since you do not describe the physics behind the Gabrovšek et al. (2018) model. It is not clear neither if you use this previous model to define the parameter of the 3D model you built.

➢ Ok we explained in more detail the physics of the model and the relation between our model and the Gabrovšek et al. (2018) model. We added the following paragraph:

*To validate the relation between the hydraulic and gravimetric signal, we have modelled both responses. A hydraulic model of the Škocjan-Kačna system was first presented by Gabrovšek et al. (2018) who demonstrated that the hydrodynamic response of a karstic system subjected to flow variations can be modelled reasonably well with simplified versions of the Navier-Stokes equations, the so-called Saint-Venant equations (SVe). With respect to the Navier-Stokes equations, the SVe model only 1-D flow, accounting the viscosity, boundary friction and turbulence terms through simple empirical relations (Blatnik et al., 2020). Due to their simplicity, such equations are particularly apt to model unsteady flow occurring in artificial channels and also in natural rivers.*

*The SVe are derived from the mass and momentum conservation of a small piece of fluid subjected to an external force. The authors used the implementation of the solution of the SVe in the Storm Water Management Model (SWMM) software developed by the Environmental Protection Agency (Rossman, 2017). SWMM is primarily designed to simulate urban sewage systems, but has been successfully applied to model conduit dominated karst systems. SWMM is an open source and versatile environment. It accounts for transitions between open channel and pressurised flow, and allows building complex conduit networks with arbitrary cross-sections. Storage is attributed to the volume of water in the conduits.*

*The input into the Škocjan-Kačna system flow is given by the discharge hydrograph of the Reka River gauging station. The geometry of the conduit system in the model is obtained from cave surveys; however, the modelling domain is highly simplified compared to reality. As shown by Gabrovšek et al. (2018), the flood behavior of the system is strongly controlled by a limited number of flow restricting conduits; furthermore, the authors demonstrated that the modelling domain should include the Kačna cave system which highly influences the water dynamics in Škocjan. The model used in the present study is an extension of the Gabrovšek et al. (2018) model in which we provide a more realistic representation of the geometry of the Škocjan Caves. Our model, in particular, increases the number of conduits in the Škocjan cave from 2 to 4 which now includes a detailed discretization of Hanke and Sumeča channels (Figure 1d).*

New reference

*Blatnik, M., Culver, D. C., Gabrovšek, F., Knez, M., Kogovšek, B., Kogovšek, J., Liu, H., Mayaud, C., Mihevc, A., Mulec, J., Năpăruş-Aljančič, M., Otoničar, B., Petrič, M., Pipan, T., Prelovšek, M., Ravbar, N., Shaw, T., Slabe, T., Šebela, S., and Zupan Hajna, N.: Deciphering Epiphreatic Conduit Geometry from Head and Flow Data, in: Karstology in the Classical Karst, edited by: Knez, M., Otoničar, B., Petrič, M., Pipan, T., and Slabe, T., Springer International Publishing, Cham, 149–168, https://doi.org/10.1007/978-3-030-26827-5_8, 2020.*

You should better emphasize the need of a 3D model to compute the gravity signal. The model you built is particularly relevant to estimate the water mass budget and you had to build

it to be able to fit the gravimetric data. To me the description of how this model functions should be integrated to the paper.

➤ Ok when explaining the calculation of the gravity effects we added the following description:

*Given the geometry of the conduits, their location in space and the simulated time-series of the water heights in the conduits, we have all the necessary information to build a 4D model of the mass variations below the gravimeter. The correct interpretation of the gravity signal induced by hydrology requires calculating the effect of the underground mass variation induced by the water flows. The model should reproduce as accurately as possible the mass distribution, including a correct spatial location of the channels, which is not required for fitting the hydraulic observations.*

*We discretized the water mass distribution occurring in the conduits at each time-step of the simulation through a series of small prisms, for which the gravity effect is calculated analytically at SK1.*

l. 244: "The final RMS difference between the observed and modelled data amounts to 8 m" → The final RMS difference between the observed and modelled **water level** data amounts to 8 m

➤ Ok added water level, thanks.

Fig. 2: Add on the c and d plots the estimate of the water level by the Gabrovšek et al. (2018) model

➤ Ok thanks, we added the estimate of water level as well as gravity as shown in reply to your first comment.

l. 350: "Detectability of water storage units in karst trough gravimetry" → Detectability of water storage units in karst **through** gravimetry

➤ Corrected, thanks for reporting this.

l. 415: "that seem to cause" → that seem**s** to cause

➤ Corrected, thanks for reporting this.

l. 422: "Apart from karst aquifers in carbonates, a similar approach can be extended to monitor cavities on gypsum and evaporates, which represent an hazard in many regions worldwide."

→ Join this lonely sentence to the above paragraph

➤ Thanks, joined to previous paragraph.

an hazard → a hazard

➤ Corrected, thanks for reporting this.

Fig. A1 a → both lines are not distinguishable since they overlap, it would be better understood if one of them ("ET model") would be represented by a dashed line. In the legend "ET model" corresponds indeed to the LTM model.

➢ Yes we agree, we re-designed the image so as to improve the representation

Fig. A1 b → in the legend the term ET is not introduced neither in the manuscript text nor in the caption, do you mean LTM?

➢ Yes you are right; by ET we mean Earth tide model, but it is better to employ LTM (local tide model) which has been already defined.

Fig. A1 c → you would ease the readability if you place the c plot below the b one

➢ Yes you are right; position of plot b is mistaken, we moved it above

Fig. A1 d → the "ET" prediction could be represented by a dashed line, change ET to LTM or specify its meaning. Explain the meaning of the terms O1, P1, K1, M2, S2, M3 or relate them to the table A1

➢ Yes we changed the colors and refer the terms to table A1.

Fig. A1 e → in the legend "No umberlla" → No umbrella

➢ Corrected, thanks for reporting this.

Fig. B1: "DEM of of the Škocjan" → DEM of the Škocjan.

➢ Corrected, thanks for reporting this.

"SK1 position" it seems that you started a sentence but didn't finish it

➢ Yes, sorry, the complete sentence is: "SK1 position shown by the red dot".

I. 653: "By 'outside' we mean the water masses contained inside the blue dashed outline in Figure B2a" → I don't see any blue dashed line or area on that plot, do you mean Fig. B1?

➢ Yes you are right; we meant figure 1B.